# WHAT DO DEEP NETS LEARN? CLASS-WISE PATTERNS REVEALED IN THE INPUT SPACE

## ABSTRACT

Deep neural networks (DNNs) have been widely adopted in different applications to achieve state-of-the-art performance. However, they are often applied as a black box with limited understanding of *what the model has learned from the data*. In this paper, we focus on image classification and propose a method to visualize and understand the class-wise patterns learned by DNNs trained under three different settings including natural, backdoored and adversarial. Different from existing class-wise deep representation visualizations, our method searches for a single predictive pattern in the input (i.e. pixel) space for each class. Based on the proposed method, we show that DNNs trained on natural (clean) data learn abstract shapes along with some texture, and backdoored models learn a small but highly predictive pattern for the backdoor target class. Interestingly, the existence of class-wise predictive patterns in the input space indicates that even DNNs trained on clean data can have backdoors, and the class-wise patterns identified by our method can be readily applied to "backdoor" attack the model. In the adversarial setting, we show that adversarially trained models learn more simplified shape patterns. Our method can serve as a useful tool to better understand DNNs trained on different datasets under different settings.

## 1 INTRODUCTION

Deep neural networks (DNNs) are a family of powerful models that have demonstrated superior learning capabilities in a wide range of applications such as image classification, object detection and natural language processing. However, DNNs are often applied as a black box with limited understanding of *what the model has learned from the data*. Existing understandings about DNNs have mostly been developed in the deep representation space or using the attention map. DNNs are known to be able to learn high quality representations (Donahue et al., 2014), and the representations are well associated with the attention map of the model on the inputs (Zhou et al., 2016; Selvaraju et al., 2016). It has also been found that DNNs trained on high resolution images like ImageNet are biased towards texture (Geirhos et al., 2019). While these works have significantly contributed to the understanding of DNNs, a method that can intuitively visualize what DNNs learn for *each class* in the *input space* (rather than the deep representation space) is still missing.

Recently, the above understandings have been challenged by the vulnerabilities of DNNs to backdoor (Szegedy et al., 2014; Goodfellow et al., 2015) and adversarial attacks (Gu et al., 2017; Chen et al., 2017). The backdoor vulnerability is believed to be caused by the preference of learning high frequency patterns (Chen et al., 2017; Liu et al., 2020; Wang et al., 2020). Nevertheless, no existing method is able to reliably reveal the backdoor patterns, even though it has been well learned into the backdoored model. Adversarial attacks can easily fool state-of-the-art DNNs by either sample-wise (Goodfellow et al., 2016) or universal (Moosavi-Dezfooli et al., 2017) adversarial perturbations. One recent explanation for the adversarial vulnerability is that, besides robust features, DNNs also learn useful (to the prediction) yet non-robust features which are sensitive to small perturbations (Ilyas et al., 2019). Adversarial training, one state-of-the-art adversarial defense method, has been shown can train DNNs to learn sample-wise robust features (Madry et al., 2018; Ilyas et al., 2019). However, it is still not clear if adversarially trained DNNs can learn a robust pattern for each class.

In this paper, we focus on image classification tasks and propose a visualization method that can reveal the pattern learned by DNNs for *each class* in the *input space*. Different from sample-wise visualization methods like attention maps, we aim to reveal the knowledge (or pattern) learned by DNNs for each class. Moreover, we reveal these patterns in the input space rather than the deep representation space. This is because input space patterns are arguably much easier to interpret. Furthermore, we are interested in a visualization method that can provide new insights into the

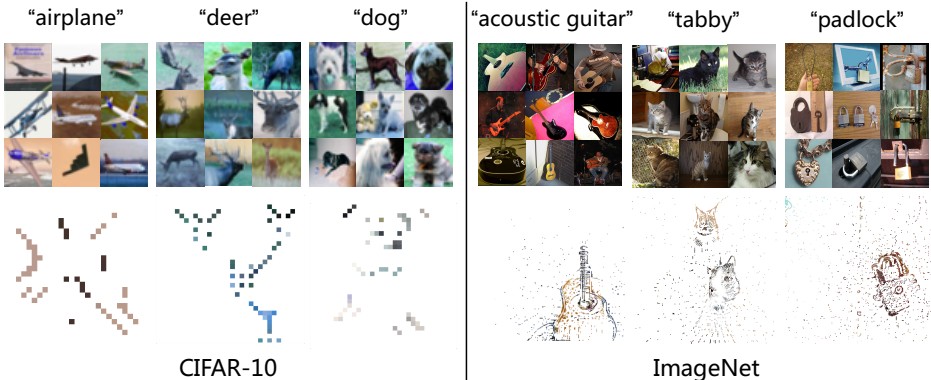

Figure 1: Example images (top row) and the class-wise patterns (bottom row) learned by a ResNet18 on CIFAR-10 (left three columns) and ResNet-50 on ImageNet (right three columns) and revealed by our method. The 3 ImageNet classes are "n02676566" ("guitar"), "n02123045"("cat"), and "n03874599" ("padlock"). The pattern size is set to 5% of the image size.

backdoor and adversarial vulnerabilities of DNNs, both of which are input space vulnerabilities (Szegedy et al., 2014; Ma et al., 2018).

Given a target class, a canvas image, and a subset of images from the nontarget classes, our method searches for a single pattern (a set of pixels) from the canvas image that is highly predictive of the target class. In other words, when the pattern is attached to images from any other (i.e. nontarget) classes, the model will consistently predict them as the target class. Figure 1 illustrates a few examples of the class-wise patterns revealed by our method for DNNs trained on natural (clean) CIFAR-10 (Krizhevsky, 2009) and ImageNet (Deng et al., 2009) datasets.

In summary, our main contributions are: 1) We propose a visualization method to reveal the class-wise patterns learned by DNNs in the input space, and show the difference to attention maps and universal adversarial perturbations. 2) With the proposed visualization method, we show that DNNs trained on natural datasets can learn a consistent and predictive pattern for each class, and the pattern contains abstract shapes along with some texture. This sheds new lights on the current texture bias understanding of DNNs. 3) When applied on backdoored DNNs, our method can reveal the trigger patterns learned by the model from the poisoned dataset. Our method can serve as an effective tool to assist the detection of backdoored models. 4) The existence of class-wise predictive patterns in the input space indicates that even DNNs trained on clean data can have backdoors, and the class-wise patterns identified by our method can be readily applied to "backdoor" attack the model. 5) By examining the patterns learned by DNNs trained in the adversarially setting, we find that adversarially trained models learn more simplified shape patterns.

## 2  RELATED WORK

**General Understandings of DNNs.** DNNs are known to learn more complex and higher quality representations than traditional models. Features learned at intermediate layers of AlexNet have been found to contain both simple patterns like lines and corners and high level shapes (Donahue et al., 2014). These features have been found crucial for the superior performance of DNNs (He et al., 2015). The exceptional representation learning capability of DNNs has also been found related to structures of the networks like depth and width (Safran & Shamir, 2017; Telgarsky, 2016). One recent work found that ImageNet-trained DNNs are biased towards texture features (Geirhos et al., 2019). Attention maps have also been used to develop better understandings of the decisions made by DNNs on a given input (Simonyan et al., 2014; Springenberg et al., 2015; Zeiler & Fergus, 2014; Gan et al., 2015). The Grad-CAM technique proposed by Selvaraju et al. (2016) utilizes input gradients to produce intuitive attention maps. Whilst these works mostly focus on deep representations or sample-wise attention, an understanding and visualization of what DNNs learn for each class in the input space is still missing from the current literature.

**Understanding Vulnerabilities of DNNs.** Recent works have found that DNNs are vulnerable to backdoor and adversarial attacks. A backdoor attack implants a backdoor trigger into a victim model by injecting the trigger into a small proportion of training data (Gu et al., 2017; Liu et al., 2018). The model trained on poisoned dataset will learn a noticeable correlation between the trigger and

a target label. A backdoored model behaves normally on clean test data, yet consistently predict a target (incorrect) label whenever the trigger appears in a test example (Zhao et al., 2020; Yao et al., 2019; Liu et al., 2020). This is believed to be caused by the fact that DNNs tend to learn more high frequency (e.g. backdoor) patterns (Chen et al., 2017; Liu et al., 2020; Wang et al., 2020). However, it is still unclear whether DNNs can learn such patterns from natural (clean) data. Moreover, despite a few attempts (Wang et al., 2019; Qiao et al., 2019), the trigger pattern still can not be reliably revealed, even though it has been well learned by the backdoored model. DNNs can also be easily fooled by small, imperceptible adversarial perturbations into making incorrect predictions (Szegedy et al., 2014; Goodfellow et al., 2016). Adversarial perturbations can be either sample-wise (Madry et al., 2018) or universal (Moosavi-Dezfooli et al., 2017). This has been found to be caused by learning useful (to prediction) but nonrobust (to adversarial perturbation) features (Ilyas et al., 2019). Meanwhile, adversarial training has been shown to learn more robust features and deliver effective defenses (Madry et al., 2018). However, existing understandings of adversarial training are established based on sample-wise attention (Ilyas et al., 2019). It still unclear, from the class-wise perspective, what robust or nonrobust input patterns look like. In this paper, we will propose a method to reveal the patterns (e.g. backdoor or adversarially robust/nonrobust) learned by DNNs for each class.

## 3 PROPOSED VISUALIZATION METHOD

In this section, we first define the input space class-wise pattern searching problem, then introduce our proposed searching method.

**Motivation and Intuition.** We focus on image classification with deep neural networks. We denote the training and test dataset as $\mathcal{D}_{train}$ and $\mathcal{D}_{test}$, respectively. Given a DNN model $f$ trained on a $K$-class $\mathcal{D}_{train}$ and a target class $y \in \{1, \cdots, K\}$, our goal is to find an input space pattern, i.e, a small set of pixels, that are extremely predictive of the target class. A highly predictive pattern of a class can largely capture the knowledge the model learned for the class. In backdoor attack, a predictive (i.e. backdoor trigger) pattern learned by the model can even control the model's prediction. Intuitively, a predictive pattern of a target class should be able to make the model consistently predict the target class whenever it is attached to images from any other (e.g. nontarget) classes.

**Class-wise Pattern Searching.** For a target class $y$, our method searches for a predictive pattern $\boldsymbol{p}_y$ from a canvas image $\boldsymbol{x}_c$, based on a small test subset $\mathcal{D}_n$ of images from the nontarget classes (i.e. $\mathcal{D}_n \subset \mathcal{D}_{test}$). The canvas image $\boldsymbol{x}_c$ is the image where the pattern (a set of pixels) is extracted. The search is done via an optimization process based on a mixed input between the canvas image $\boldsymbol{x}_c$ and an image $\boldsymbol{x}_n \in \mathcal{D}_n$. The mixed input $\tilde{\boldsymbol{x}}$ is defined as follows:

$$\tilde{\boldsymbol{x}} = \boldsymbol{m} * \boldsymbol{x}_c + (1 - \boldsymbol{m}) * \boldsymbol{x}_n, \tag{1}$$

where $\boldsymbol{m}$ is a mask that has the same size as either $\boldsymbol{x}_c$ or $\boldsymbol{x}_n$, and $\boldsymbol{m}_{ij} \geq 0$. The mixed input image is labeled as the target class $y$ regardless of its original class. This mixing strategy is reminiscent of the mixup (Zhang et al., 2018) data augmentation algorithm. However, we do not mix the class labels and our purpose is for pattern optimization rather than data augmentation.

During the searching process, the mask is iteratively updated to minimize the following loss:

$$\mathcal{L} = -\log f_y(\tilde{\boldsymbol{x}}) + \alpha \frac{1}{n} \|\boldsymbol{m}\|_1, \tag{2}$$

where, $f_y$ is network's probability output with respect to target class $y$, $\|\cdot\|_1$ is the $L_1$ norm, $\alpha$ is a parameter that balances the two loss terms, and $n$ is the size of the input image as well as the mask. The first loss term is the commonly used cross entropy loss. The second term increases the sparsity of mask as we are interested in simple patterns with a small number of highly predictive pixels.

During the search process, we pair the canvas image $\boldsymbol{x}_c$ randomly with images from $\mathcal{D}_n$, and iteratively update the mask $\boldsymbol{m}$ using standard Stochastic Gradient Decent (SGD) while keeping the model parameters unchanged. At each iteration, the mask $\boldsymbol{m}$ will also be clipped into $[0, 1]$. Once a mask is learned, we further clip the values in the mask that are smaller than $\gamma$ to zero, larger than $\gamma$ to one. We denote this clipped mask by $\boldsymbol{m}_\gamma$. We then extract the pattern from the canvas image by $\boldsymbol{p}_y = \boldsymbol{m}_\gamma * \boldsymbol{x}_c$. The $\gamma$ parameter can be flexibly determined in different applications. A large $\gamma$ may lead to less predictive pattern while a small $\gamma$ will produce more of a sample-wise pattern that overfits to the canvas image.

The above search method is repeatedly applied to $N$ canvas images to generate $N$ patterns for each class. We then select the pattern that has the lowest loss value as the final pattern of the class. This

additional step is to find the most predictive pattern by exploring different canvases. The complete procedure of our method is described in Algorithm 1 in Appendix A.

**Canvas Sampling.** We propose four different sampling strategies for the selection of the $N$ canvas images: positive sampling, negative sampling, random sampling and white canvas.

Positive sampling selects the top-$N$ confident images from the target class according to the logits output of model $f$. Negative sampling selects the top-$N$ most confidently misclassified images from any nontarget class into the target class. The random sampling randomly chooses $N$ images from the target class $y$. The white canvas simply uses an image with all white pixels as the canvas.

Both positive and the negative sampling aim to find the most well-learned examples by the model, but from different perspectives: well-learned correctly (e.g. positive) vs. well-learned incorrectly (e.g. negative). The white canvas is interesting since the pattern found from the white canvas will have the texture "removed", which is useful for scenarios where only the shape features are of interest. The patterns found based on different canvases are compared in Figure 4. After applying our method on each class, we can obtain a set of class-wise patterns: $\mathcal{P} = \{\boldsymbol{p}_1, \cdots, \boldsymbol{p}_K\}$. This set of predictive patterns can revel the knowledge learned by model $f$ for each class from a unique perspective.

**Why is it Class-wise?** At first sight, one might wonder if the discovered pattern could be sample-wise, rather than class-wise, given the use of the canvas sample. Note that, however, even though we are using a single sample as a canvas, the pattern found by the optimization algorithm is dependent on how the model has learnt the entire class, in terms of its loss. This is particularly evident in the case of the all white canvas, which bears no relation to any input sample. Hence our designation of the pattern as being "class-wise". While our method can find consistent and predictive class-wise patterns in the experiments, it might still be extendable. For example, using multiple positive canvas images at the same time, using noise rather than the non-target images, or using universal adversarial perturbation (UAP) (Moosavi-Dezfooli et al., 2017) but in a more controlled manner. We leave further explorations of these methods as our future work.

**Difference to Universal Adversarial Perturbation.** UAP can also be applied to craft class-wise adversarial patterns that can make the model predict an adversarial target class. In this view, both UAP and our method find predictive patterns to the target class. However, the two methods work in different ways. By fooling the network, UAP explores the *unlearned* space (low-probability "pockets") of the network (Szegedy et al., 2014; Ma et al., 2018). In contrast, our method is a searching (rather than perturbing) method that does not rely on adversarial perturbations. Thus, it has to find the optimal pixel locations in the input space that are *well-learned* by the model for the pattern to be predictive of the class. In Section 4.2 and Appendix E, we have experiments showing the difference of the patterns found by class-wise UAP and our method.

## 4 EXPERIMENTS

In this section, we will apply our method to reveal the class-wise patterns learned by DNNs trained on natural data, backdoored data, adversarial settings with intentionally-perturbed robust or nonrobust datasets, and adversarially trained DNNs. We also conduct comparisons with attention maps and universal adversarial perturbations.

**Experimental Setting.** We consider ResNet-50 (He et al., 2016) on two benchmark image datasets: CIFAR-10 (Krizhevsky, 2009) and ImageNet (Deng et al., 2009). The models are trained on the training set of the datasets using standard training strategies, except the adversarial experiments in Section 4.4. We then apply our method to search for the class-wise patterns based on canvas images and nontarget-class images selected from the test sets of the datasets. Here, we set the nontarget-class subset to be 20% of the test images (more analysis of the nontarget-class subset size can be found in Appendix C.2). The sparsity regularization parameter $\alpha$ is slightly adjusted around 0.2 for different classes and datasets. The clipping parameter $\gamma$ is set to discover a pattern of a predefined size (e.g. 5% or 10% of the image size), which will be explicitly stated in each experiment. The searching process is run for 5 epochs with respect to the nontarget-class subset. All experiments are run on GeForce GTX 1080Ti GPUs.

**Predictive Power.** We define the predictive power of the pattern $\boldsymbol{p}_y$ for target class $y$ as follows: $PW = \frac{ACC(f(\boldsymbol{x}_n + \boldsymbol{p}_y), y)}{ACC(f(\boldsymbol{x}_y), y)}$, where $ACC(f(\boldsymbol{x}_n + \boldsymbol{p}_y), y)$ is the model's accuracy on nontarget images $\boldsymbol{x}_n \in \mathcal{D}_{test}$ when attached with the target-class pattern $\boldsymbol{p}_y$ with respect to label $y$, and $ACC(f(\boldsymbol{x}_y), y)$ is the model's original accuracy on the target class images and $\boldsymbol{x}_y \in \mathcal{D}_{test}$. Intu-

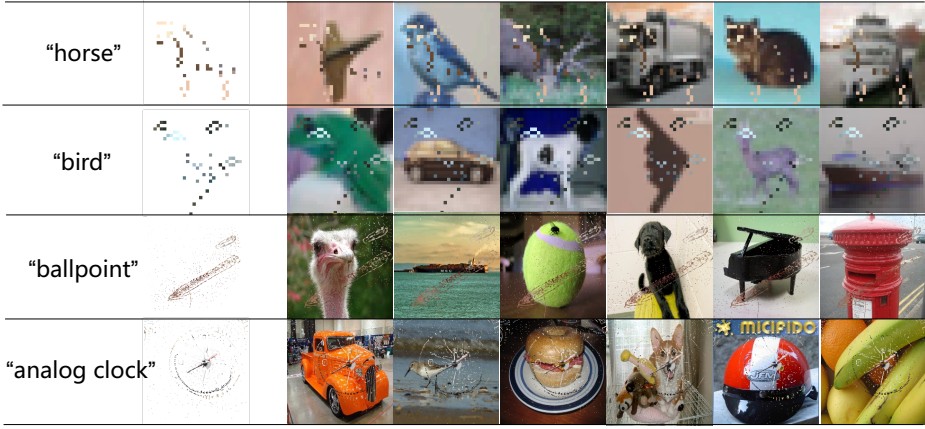

Figure 2: Class-wise patterns (second column) learned by a ResNet-50 on natural CIFAR-10 (top two rows) and ImageNet (bottom two rows). Class names are shown in the first column, while nontarget-class images with the pattern attached and also being misclassified as the target class are shown in columns 3-8. The 2 ImageNet classes are "n02783161"("ballpoint") and "n02708093"("analog clock"). Pattern size is 5% of the image size.

itively, the $PW$ metric reflects the degree to which the pattern can determine the model's prediction towards the target class, relative to the true target-class images. Note that, in most of our experiments, the pattern $\boldsymbol{p}_y$ is searched on $\mathcal{D}_n \subset \mathcal{D}_{test}$ which is obtained from 20% of the test set. And the PW metric is computed on the entire (i.e. 100%) test set. We have an experiments in Appendix B.2 showing consistent results of PW on the 80% test set that has not been used for pattern searching.

## 4.1 CLASS-WISE PATTERNS LEARNED BY DNNs FROM NATURAL DATA

We first show the class-wise patterns learned by ResNet-50 on natural CIFAR-10 and ImageNet datasets. Here, we use positive canvases sampled using the positive sampling (see Section 3). A more detailed analysis on the impact of different positive canvases can be found in Appendix D.3. The patterns revealed at 5% image size for two CIFAR-10 and ImageNet classes are illustrated in Figure 2. More patterns for all 10 CIFAR-10 classes and 20 ImageNet classes are provided in Figure 9 and 10 in Appendix C.1. The predictive power of different sizes (0% - 10%) of patterns are shown in Figure 3. The detailed predictive power results for each of the 10 CIFAR-10 classes and 50 randomly selected ImageNet classes can be found in Appendix B.1.

**Patterns Revealed by Positive Canvas.** As shown in Figure 2, the class-wise patterns learned from natural data contain shapes that are closely associated with the object class, although some of the shapes are quite abstract, especially for the CIFAR-10 classes. Interestingly, DNN can learn more than one shape for a given class, for example the "tabby" class in Figure 1 and "ballpoint" class in Figure 2. And the shapes can be found at different locations of the input space (e.g. "padlock" in Figure 1). This phenomenon is more common on ImageNet, as shown in Figure 10. This is because even the same class of objects may have different appearances and appear at different locations of the image. As a result, the model learns to be sensitive to more than one patterns at different locations of the input space. According to the mean predictive power shown in Figure 3 (left subfigure), the patterns revealed by our method represent roughly 60% of the predictive information learned by the model. And the predictive power transfers well across separately trained models, as shown in the right subfigure of Figure 7. This indicates that DNNs can learn to capture some common shapes for each class, even though the training images in the same class might be quite diverse.

The abstract shapes are not the whole story here as there are also some textures (e.g. color) in the pattern. Shapes are more class-wise features as our method iteratively searches for the optimal positions of the pixels that are predictive to the entire class. By contrast, the textures are more sample-wise features carried over from the canvas image by the $\boldsymbol{p}_y = \boldsymbol{m}_\gamma * \boldsymbol{x}_c$ operation. The use of a white-canvas can remove sample-wise textures while keeping only class-wise shapes (see Figure 4). In relation to existing texture bias understanding of DNNs trained on ImageNet (Geirhos et al., 2019), our result implies that shape also plays an important role in DNNs. Note that the $L1$ regularization of our method does not restrict the pixels in the pattern to be texture or shape. The lack of texture in the patterns indicates that DNNs do learn to be more sensitive to certain positions of the

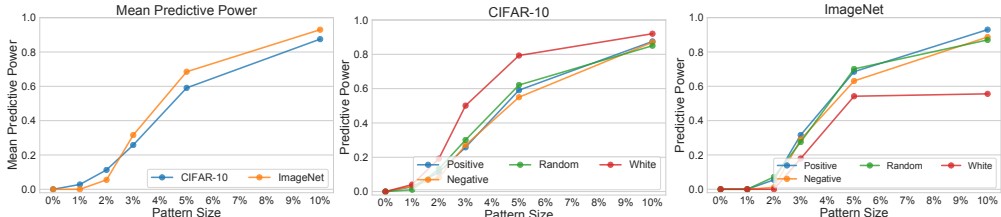

Figure 3: *Left*: mean predictive power of the class-wise patterns of different sizes, over all 10 CIFAR-10 classes and 50 randomly selected ImageNet classes. *Middle*: mean predictive power of the patterns found with different types of canvases on CIFAR-10. *Right*: mean predictive power of the patterns found with different types of canvases on ImageNet. The patterns are searched on ResNet-50 models trained on clean CIFAR-10 and ImageNet.

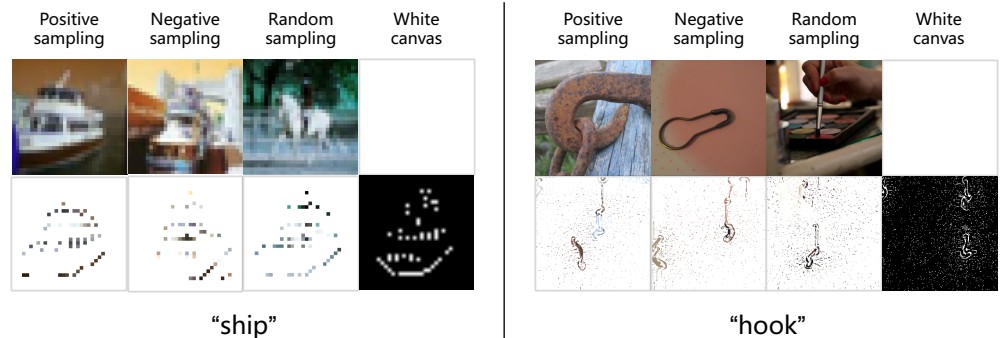

Figure 4: Patterns (second row) found by our method on different types of canvases (the first row) for CIFAR-10 "ship" class (left) and ImageNet "hook"("n03532672") class (right).

input space. Note that, besides shape and texture, DNNs also learn a certain amount of background noise, especially on ImageNet.

**Patterns Revealed by Different Types of Canvases.** An example of the patterns found by the 4 types of canvases are illustrated in Figure 4. More visualizations are provided in Figure 13 (CIFAR-10) and Figure 14 (ImageNet) in Appendix D.2. As can be observed, the abstract shapes found on different canvases are all closely associated with the object class, although there are some variations in the details and locations. As shown in Figure 14, for some of the canvases, the revealed pattern has a noticeable correlation with the canvas, especially for the positive canvas. For example, the first two patterns for the "analog clock" class (the second row in Figure 14). This implies that the patterns found by our method do carry sample-wise features from the canvas image, though they are very predictive of the entire class. Note that the patterns revealed by the white canvas also contain clear shapes, though there are no sample-wise features. More analysis of the impact of different canvas on the predictive power can be found in Appendix D.1 and Appendix D.3. While the *mean* (over all target classes) predictive powers are similar for patterns found on different canvases (Figure 12), the individual predictive power varies across different canvas images (Table 4 and Table 5). This is also why we suggest to use $N$ (i.e. $N$=5) different canvases to help find the most predictive patterns.

### 4.2 COMPARISON TO ATTENTION MAP AND UNIVERSAL ADVERSARIAL PERTURBATION

**Comparison to Attention Map.** Here, we compare the predictive power of the class-wise patterns found by our method to the key areas identified by using the attention map. This is somewhat not a fair comparison since attention maps were designed for sample-wise explanations. However, we believe it is interesting to see if high attention areas are also predictive of the class. We use Grad-CAM (Selvaraju et al., 2016) to extract the high attention area of the positively sampled canvas image for each of the 50 ImageNet target classes (class names are in Table 2). The size of the high attention area is set to be 10% of the image. We then attach the extracted attention area to all non-target class test images to compute its predictive power (the same testing scheme as for our method). Figure 5 illustrates a few example attention areas (left grid) and the mean predictive power over the 50 classes (right plot). We find that the predictive power of the attention patterns are considerably low. This indicates that sample-wise attention maps may not be a good representation of the class-wise knowledge learned by the model. As the attention visualization shown in Appendix

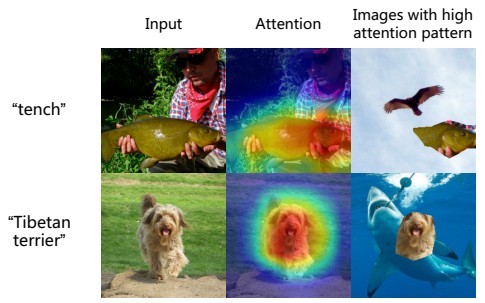 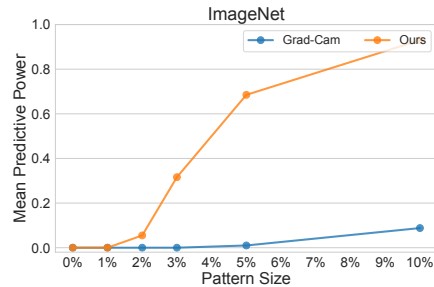

Figure 5: *Left grid*: input image (first column), attention map (second column) and images (third column) attached with high attention patterns extracted from the input image. *Right plot*: the mean predictive power of high attention patterns over 50 ImageNet target classes. The attention pattern size is 10% of the image size.

H, the patterns identified by our method indeed have a noticeable correlation to the class and can consistently attract the network's attention when attached to different images.

**Comparison to Universal Adversarial Perturbations.** Here we visualize the patterns revealed by UAP, our method and one variant of our method with the total-variation (TV) regularization (Fong & Vedaldi, 2017). We apply UAP targeted attack in a class-wise fashion, accumulating the perturbations separately for each target class. The class-wise UAP is crafted on the entire test set for CIFAR-10 and 10K randomly selected test images for ImageNet. The loss function used by the TV variant of our method is defined as follows:

$$\mathcal{L} = -\log f_y(\tilde{\boldsymbol{x}}) + \alpha \frac{1}{n} \|\boldsymbol{m}\|_1 + \beta \frac{1}{n} \|\nabla \boldsymbol{m}\|_2^2 \,, \tag{3}$$

where $\beta$ is a parameter that controls the strength of the TV regularization (the third term). The reason we consider the TV regularization is that it can reduce the variation of the mask, producing more smooth patterns (Fong & Vedaldi, 2017).

As shown in Figure 6, the UAP patterns generated on CIFAR-10 also contain structures of the object. However, the patterns found by our method are much cleaner, although there are still some background noises. The UAP patterns generated on ImageNet are notably more abstract than our methods. This is because, by fooling the network, UAP explores the *unlearned* space of DNNs. By contrast, our method is a searching (rather than perturbing) method that does not rely on adversarial perturbations. Therefore, it needs to find the *well-learned* locations in the input space for the pattern to be predictive of the class. We believe UAP can be applied in a more controlled manner to explore the well-learned (rather than unlearned) space of the network. We find that the TV regularization can indeed help produce smooth patterns with less background noise, however, the main patterns are generally the same as those found without using TV. More visualizations and a predictive power analysis of UAP and our methods can be founded in Appendix E.

### 4.3 CLASS-WISE PATTERNS LEARNED BY BACKDOORED DNNS

Here, we apply our method to reveal the patterns learned by a backdoored ResNet-50 model by BadNets (Gu et al., 2017) on CIFAR-10. The model was trained without data augmentation (see Appendix F for the results with data augmentations). The patterns revealed on different canvases are illustrated in the left figure of Figure 7. Here, we set the pattern size to be small (e.g. 1%) as here we are interested in examining whether the model has learned a small, extremely predictive but meaningless pattern. As can be observed, our method can reliably reveal the trigger pattern, especially on the white canvas. More visualizations with three types of backdoor triggers can be found in Figure 16 in Appendix F.

We then test the predictive power of the recovered trigger patterns on the backdoored model. For a comparison, we also test the predictive power of the same size of pattern identified on clean model. Here, we apply BadNets to attack all CIFAR-10 classes individually, and compute the mean predictive power over the 10 classes. As shown in the middle figure of Figure 7, the patterns are highly predictive on the backdoored model, yet the same size of pattern identified on natural has almost zero predictive power on the natural model itself. This confirms that the model indeed learns to memorize the trigger pattern. On the other hand, the existence of small but extremely predictive

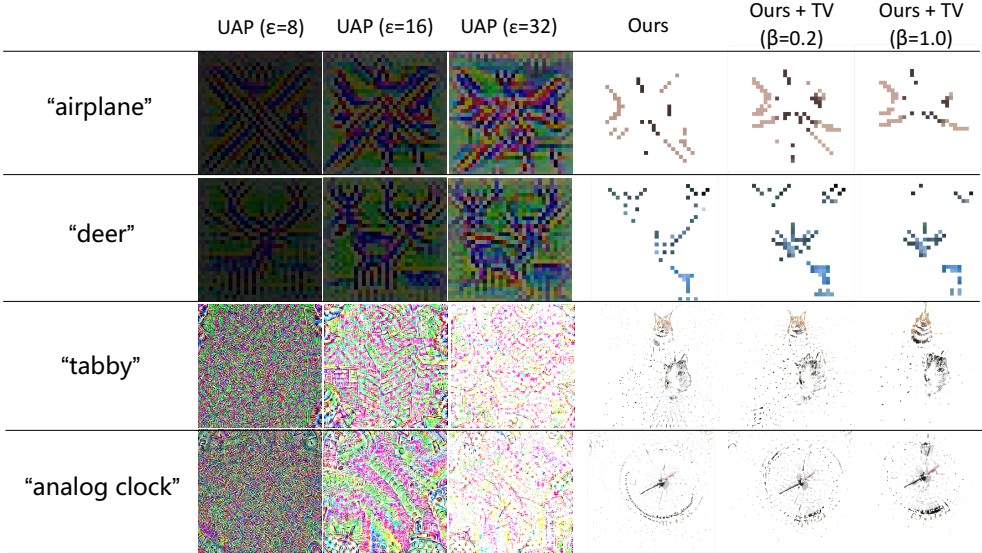

Figure 6: A comparison of the class-wise patterns revealed on naturally trained ResNet-50 models by universal adversarial perturbation (UAP), our method and our method with the total-variation (TV) regularization (defined in Equation 3). For our methods, the pattern size is set to 5% of the image size, while UAP perturbs the entire (i.e. 100%) image. The top and bottom two rows show the patterns for two CIFAR-10 and ImageNet classes, respectively.

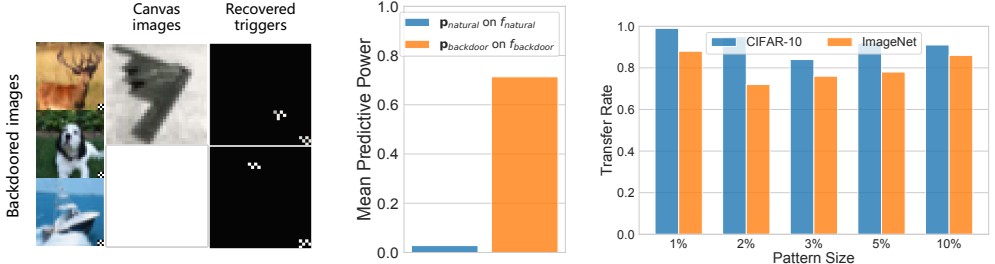

Figure 7: *Left*: Backdoor trigger patterns recovered by our method from a backdoored (by BadNets with target class "airplane") ResNet-50 model on CIFAR-10 using different canvas images.The trigger pattern is a black-white checkerboard at the bottom right corner of the image. *Middle*: The mean (over all 10 CIFAR-10 backdoor target classes) predictive power of 1) the recovered trigger pattern on the backdoored model ($p_{backdoor}$ on $f_{backdoor}$), and 2) the same size of natural pattern on natural model ($p_{natural}$ on $f_{natural}$). *Right*: The transferability of the patterns revealed for clean ResNet-50 models. The transfer rate is defined as $PW_{source}/PW_{target}$, the ratio between the predictive power on the source and the target model. The patterns are searched on the source model.

patterns can be used to assist the detection of backdoored models. The effectiveness of our method on more complex backdoor triggers is worth further exploration.

Revisiting the patterns learned on natural data (Figure 2), we find that DNNs trained on natural data can also have backdoors, since those patterns can be immediately applied to "backdoor" attack the model. The images shown in columns 3-8 in Figure 2 are the backdoored images. The predictive powers shown in Figure 3 indicate that the attack success rate will be high, although not as high as the state-of-the-art (Chen et al., 2017; Liu et al., 2020) and stealthiness is not guaranteed. Interestingly, these results indicate that DNNs tend to memorize certain patterns for each class, which explains why backdoor attacks can easily succeed by poisoning only a small proportion of the training data.

### 4.4 CLASS-WISE PATTERNS LEARNED BY DNNs IN ADVERSARIAL SETTINGS

Here, we apply our method to identify the class-wise patterns learned by ResNet-50 on three different versions of CIFAR-10 dataset: natural ($\mathcal{D}$), robust ($\mathcal{D}_R$) and nonrobust ($\mathcal{D}_{NR}$). The $\mathcal{D}_R$ and $\mathcal{D}_{NR}$ versions were generated in previous work (Ilyas et al., 2019) by perturbing the training images

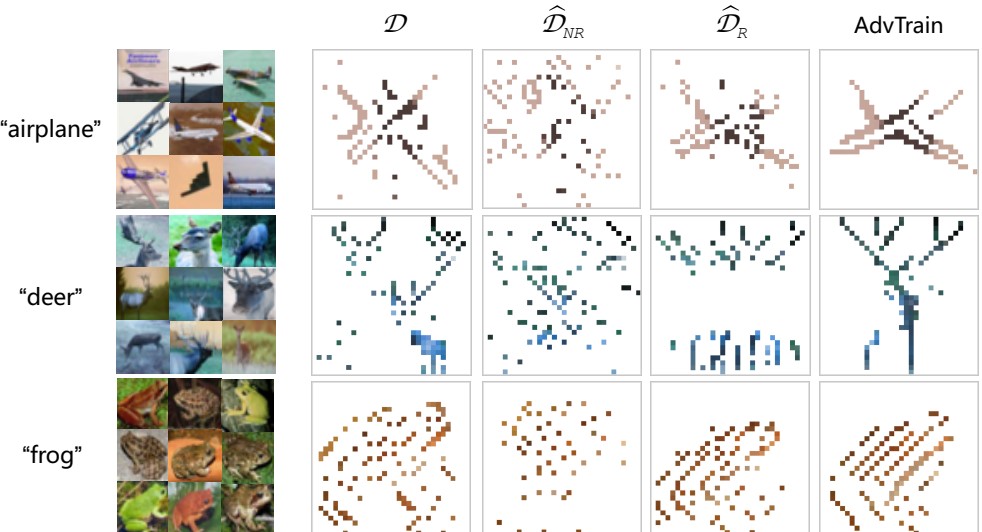

Figure 8: Examples of class-wise patterns learned by ResNet-50 and revealed by our method on natural ($\mathcal{D}$), nonrobust ($\mathcal{D}_{NR}$) and robust ($\mathcal{D}_R$) data on CIFAR-10, as well as adversarially trained model (AdvTrain) on CIFAR-10. Pattern size is 10% of the image size.

of the original dataset $\mathcal{D}$ to have only robust or nonrobust features in an adversarial setting. In other words, $\mathcal{D}_R$ contains only the robust features learned by a robust (adversarially trained) model, while $\mathcal{D}_{NR}$ contains only the nonrobust features learned by a nonrobust model. It was shown that DNNs exhibit moderate adversarial robustness when trained on $\mathcal{D}_R$ using even standard training (Ilyas et al., 2019). We train the ResNet-50 model independently using standard training on $\mathcal{D}$, $\mathcal{D}_{NR}$ and $\mathcal{D}_R$, and adversarial training on $\mathcal{D}$.

The class-wise patterns learned under different settings are illustrated in Figure 8. Compared to $\mathcal{D}$, the patterns learned on $\mathcal{D}_{NR}$ are mostly background noise. However, patterns learned by the adversarially trained model contains clear shapes and much less background noise. Compared to that learned on $\mathcal{D}$, the shapes learned by adversarially trained DNNs are more simplified and have more connected regions. This provides an interesting visualization of the robust patterns learned by adversarially trained DNNs from a class-wise perspective. The robust dataset $\mathcal{D}_R$ is interesting as it is a dataset that is intentionally perturbed to be robust. The patterns learned on $\mathcal{D}_R$ reveal that the "robust" perturbation can remove much of the background noise (e.g. the "airplane" class). However, it may also remove a certain part of the robust shape (e.g. the body of the "deer"). This might produce DNNs that have poor generalization performance in real-world scenarios.

## 5  DISCUSSION AND CONCLUSION

In this paper, we propose a method to identify the *class-wise* patterns learned by deep neural networks (DNN) and reveal the patterns in the *input space*. The patterns revealed by our method indicate that DNNs can learn abstract but consistent shape patterns that are associated with the target class. By further examining the patterns learned by backdoored DNNs, we find that our method can reveal the trigger pattern, and more interestingly, DNNs trained on clean data may also have "backdoors". Patterns revealed in adversarial settings indicate that adversarially trained DNNs can indeed learn more robust shape patterns, however, internationally-perturbed robust datasets may lose certain shape features. Our method can serve as a useful tool for communities to better understand DNNs trained on different datasets under different settings.

One application of our method is DNN understanding and interpretation. Based on our method, one can develop metrics to measure the strength, weakness or biases (e.g. gender, color and other attributes) of the knowledge learned by DNNs. The other potential application is using our method to develop effective backdoor defense methods by monitoring and avoiding the learning of high predictive patterns during training. Our method can also motivate more effective adversarial defense methods, for example, regularized adversarial training methods that can help DNNs learn more robust shape features.

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

# A  ALGORITHM AND SEARCH SETTINGS

The proposed Input Space Class-wise Pattern (ISCP) searching method is described in Algorithm 1. Given a DNN model $f$ (trained on $\mathcal{D}_{train}$), a target class $y$ and a canvas image $\boldsymbol{x}_c$ (in the case of positive and negative canvas, $\boldsymbol{x}_c$ is sampled from $\mathcal{D}_{test}$), the pattern search produce is executed as follows. The mask $\boldsymbol{m}$ is initialize to have the same size with the input image (without the channel dimension) and all 0.5 values (line 1). We randomly sample a subset of non-target class images $\mathcal{D}_n$ from the test set $\mathcal{D}_{test}$, which in most of our experiments, is set to the non-target class images from 20% of the test set.

The searching procedure iterates for $T = 5$ epochs ($T$ times full pass on $\mathcal{D}_n$) with minibatch size $B = 4$. In each epoch, for each minibatch $\{\boldsymbol{x}_n^{(j)}\}_{j=1}^B$ in $\mathcal{D}_n$, we first mix each image in the minibatch with the canvas image $\boldsymbol{x}_c$ using the mask $\boldsymbol{m}$ (line 4). We then compute the loss on the mixed inputs (line 5), and backpropagate to update the mask according to a step size $\eta = 0.02$ (lines 6-7). The updated mask is then clipped into the value range of $[0, 1]$.

After $T$ epochs of optimization, we obtain a mask $\boldsymbol{m}$. We then clip the values according to a threshold $\gamma$, so that only a certain proportion (e.g. 5%) of the values are clipped to 1 (the rest are clipped to 0). We then apply the clipped mask on the canvas image $\boldsymbol{x}_c$ to obtain the final pattern $\boldsymbol{p}_y$ for target class $y$. The loss term balancing parameter $\alpha \in [0.1, 0.3]$ is slightly adjusted around 0.2 to produce the most predictive pattern. The $\gamma$ parameter is set to the value that only a certain percentage of the pixels are remained, which in most of our experiments is 5% of the image size.

---

**Algorithm 1** Input Space Class-wise Pattern (ISCP) Searching Method

---

**Require:**
   A DNN model $f$, canvas image $\boldsymbol{x}_c$, nontarget-classes subset $\mathcal{D}_n \subset \mathcal{D}_{test}$, target class $y$, regularization balancing parameter $\alpha$, clipping threshold $\gamma$, input image size $W \times H$, epochs $T = 5$, batch size $B = 4$, step size $\eta = 0.02$.
**Ensure:** Pattern $\boldsymbol{p}_y$ for target class $y$
 1: $\boldsymbol{m} =$ InitializeMask()
 2: **for** $t$ in range($T$) **do**
 3:     **for** $\{\boldsymbol{x}_n^{(j)}\}_{j=1}^B$ in $\mathcal{D}_n$ **do**
 4:         $\tilde{\boldsymbol{x}}^{(j)} = \boldsymbol{m} * \boldsymbol{x}_c + (1 - \boldsymbol{m}) * \boldsymbol{x}_n^{(j)}$
 5:         $\mathcal{L} = \sum_{j=1}^B \{-\log f_y(\tilde{\boldsymbol{x}}^{(j)}) + \alpha \frac{1}{n} \|\boldsymbol{m}\|_1\}$
 6:         $\boldsymbol{\delta} = \frac{\partial \mathcal{L}}{\partial \boldsymbol{m}}$
 7:         $\boldsymbol{m} = \boldsymbol{m} - \eta * \text{sign}(\boldsymbol{\delta})$
 8:         Clip all the values in $\boldsymbol{m}$ to $[0, 1]$
 9:     **end for**
10: **end for**
11: Clip the values in $\boldsymbol{m}$ that are smaller than $\gamma$ to 0, larger than $\gamma$ to 1, and get $\boldsymbol{m}_\gamma$
12: $\boldsymbol{p}_y = \boldsymbol{m}_\gamma * \boldsymbol{x}_c$
13: return $\boldsymbol{p}_y$

---

# B  MORE PREDICTIVE POWER RESULTS

## B.1  PREDICTIVE POWER FOR EACH CLASS

Here, we show the detailed predictive powers of the class-wise patterns revealed by our method for all 10 CIFAR-10 classes and 50 randomly selected ImageNet classes. The experimental settings follows Section 4.1: ResNet-50 models trained on natural CIFAR-10 and ImageNet. The patterns are searched based on 20% of the test images, and the predictive power (defined in Section 4) is computed on all (e.g. 100%) test images. In Figure 3, we have shown the mean predictive power over all 10 classes of CIFAR-10 and 50 classes of ImageNet, here we show the detailed predictive power for each class in Table 1 and Table 2. Note that our method only searches one pattern for each class, and the pattern size is set to be 1% - 10% of the image size. Positive sampling of the canvases are used in these experiments.

As shown in Table 1 and 2, the predictive power is extremely low when the pattern size is below 5% of the image size. This confirms our findings in the backdoor trigger recovery experiment in Section 4.3: on naturally trained models, small patterns that are of 1% the image size are not

Table 1: Detailed predictive power of different sizes of the patterns found by our method for ResNet-50 trained on natural CIFAR-10. The predictive power is shown separately for each CIFAR-10 class. **mPW**: mean predictive over all classes; **STD**: standard deviation over all classes. Positive canvases are used.

| Class | Pattern Size | | | | | Original Acc |
|---|---|---|---|---|---|---|
| | 1% | 2% | 3% | 5% | 10% | |
| airplane | 0.031 | 0.126 | 0.357 | 0.745 | 0.892 | 0.953 |
| automobile | 0.032 | 0.081 | 0.246 | 0.663 | 0.921 | 0.979 |
| bird | 0.043 | 0.161 | 0.333 | 0.665 | 0.955 | 0.932 |
| cat | 0.047 | 0.077 | 0.162 | 0.431 | 0.814 | 0.851 |
| dear | 0.021 | 0.104 | 0.365 | 0.698 | 0.771 | 0.960 |
| dog | 0.011 | 0.099 | 0.270 | 0.571 | 0.867 | 0.910 |
| frog | 0.010 | 0.041 | 0.104 | 0.383 | 0.838 | 0.967 |
| horse | 0.040 | 0.151 | 0.278 | 0.596 | 0.865 | 0.948 |
| ship | 0.010 | 0.124 | 0.248 | 0.589 | 0.901 | 0.968 |
| truck | 0.032 | 0.167 | 0.212 | 0.570 | 0.928 | 0.957 |
| **mPW** | 0.028 | 0.113 | 0.258 | 0.591 | 0.875 | 0.942 |
| **STD** | 0.013 | 0.038 | 0.079 | 0.107 | 0.053 | 0.036 |

predictive unless the model has learned a backdoor trigger. At pattern size 5%, the predictive power on CIFAR-10 is 0.59, which means our method has found a pattern that can represent 59% of the original accuracy on the target class. While these patterns exhibit different predictive power on different classes, the variation is fairly low. This indicates that our method can found representative and predictive patterns reliably for different classes. At pattern size 10%, the mean predictive power across all 10 CIFAR-10 classes is 0.875, that is, the pattern can represent 87.5% of the original accuracy on average. Large patterns are generally more predictive, but contains more background noise. As shown in Table 2, the results on ImageNet are similar to that on CIFAR-10, except that for the same pattern size, the patterns are more predictive on ImageNet and the variation is higher. This is because ImageNet has more diverse images in the same class than CIFAR-10.

### B.2 PREDICTIVE POWER ON DIFFERENT TEST SETS

In our setting, the patterns are searched with the assistance of 20% clean test data, after which the patterns are applied to the entire (i.e. 100%) test set to compute their predictive powers. Here, we test the predictive power on the rest of the 80% test data that were not used for pattern search. This is to rule out the influence of the 20% searching subset on the predictive power. We compute the predictive power for naturally trained ResNet-50 models on CIFAR-10 and ImageNet, and take the mean predictive power over all 10 classes of CIFAR-10 and 50 randomly selected ImageNet classes. We also vary the pattern size from 1% to 10% of the image size to visualize the trend.

The results are visualized in Table 3. As can be inferred, the mean predictive powers tested on the entire test set are very close to that tested on the 80% test set that were not used for pattern searching. This confirms that the patterns identified by our methods remain predictive on unseen images. A formal transferability study of our pattern across separately trained models can be found in Figure 7 (right panel).

## C MORE VISUALIZATIONS OF THE CLASS-WISE PATTERNS

### C.1 CLASS-WISE PATTERNS FOR MORE CIFAR-10 AND IMAGENET CLASSES

Figure 9 illustrates the patterns found by our method for each of the 10 CIFAR-10 classes, while Figure 10 shows the patterns for 20 out of the 50 selected ImageNet classes. As can be observed in Figure 9, our method can find interpretable patterns for all CIFAR-10 classes, and the patterns generally contain shape features that well describe the object class. The color of the pattern reflects the texture information learned by the model to some extent. Comparing the patterns found on CIFAR-10 to that on ImageNet in Figure 10, DNNs learn more detailed shapes on high resolution images. Another observation is that DNNs can learn more than one shapes at different locations of the input space, especially on ImageNet. The patterns revealed by our method seem to have a noticeable correlation with the high occurrence objects of the class.

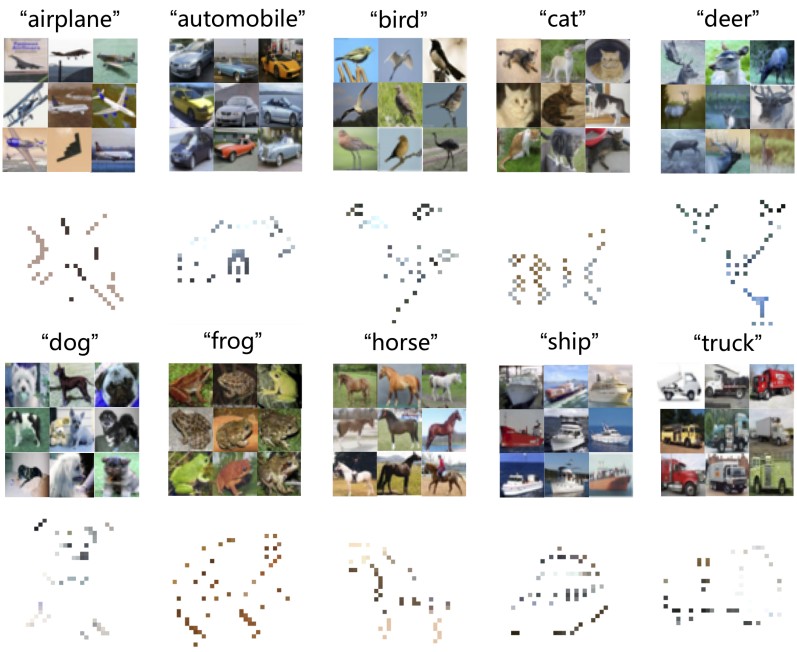

Figure 9: Class-wise patterns found by our method on CIFAR-10 at pattern size 5%. We show the pattern for each of the 10 CIFAR-10 classes. The predictive powers of these patterns are reported in Table 1. Positive canvases are used.

## C.2 Patterns found with Different Sizes of the Nontarget-class Subset

Here, we show the patterns revealed by our method with different sizes of the nontarget-class subset selected from the original test set. We apply our method to search for the patterns with different nontarget-class subsets of sizes ranging from 20% to 100% of the test set. Note that the predictive powers of the patterns found in each setting are still computed on the entire (i.e. 100%) test set. This experiment is conducted on a ResNet-50 model trained on natural CIFAR-10 dataset, and the pattern size is fixed to 5% of the image size. The class-wise patterns and their predictive powers are illustrated in Figure 11. As the results show, the patterns are fairly consistent under different subset sizes (left figure), and 20% of the test set is sufficient for our method to find highly predictive patterns (right figure).

## D    More analyses of the canvases

### D.1    Effect of different canvases on predictive power

Here, we present more detailed results of the effect of the canvas type on the predictive power. The four types of canvases (e.g. random, positive, negative and white) are described in Section 3. The experiments are conducted with naturally trained ResNet-50 models on CIFAR-10 and ImageNet. The patterns are searched on 20% of the test images and the predictive power is computed on all (e.g. 100%) of test images. The search setting of our method follows that described in Section 4. For CIFAR-10, we compute the mean predictive power (mPW) over all 10 classes, while for ImageNet, we compute the mPW on 50 randomly selected classes.

Figure 12 illustrate the mPW on CIFAR-10 and ImageNet. We find that patterns found on random and negative canvases are similarly predictive of the class. An interesting observation is that, on CIFAR-10, patterns found on the white canvas have even higher predictive power than those found on other canvases. However, the white canvas is less effective on ImageNet. We suspect this is because CIFAR-10 images are all low-resolution images, on which the model tends to learn patterns that can be well represented by one single type of pixels. On high-resolution ImageNet images, the model learns more complex patterns that are associated with the textures. In this case, white pixels become less effective to represent the high-resolution patterns learned by the model.

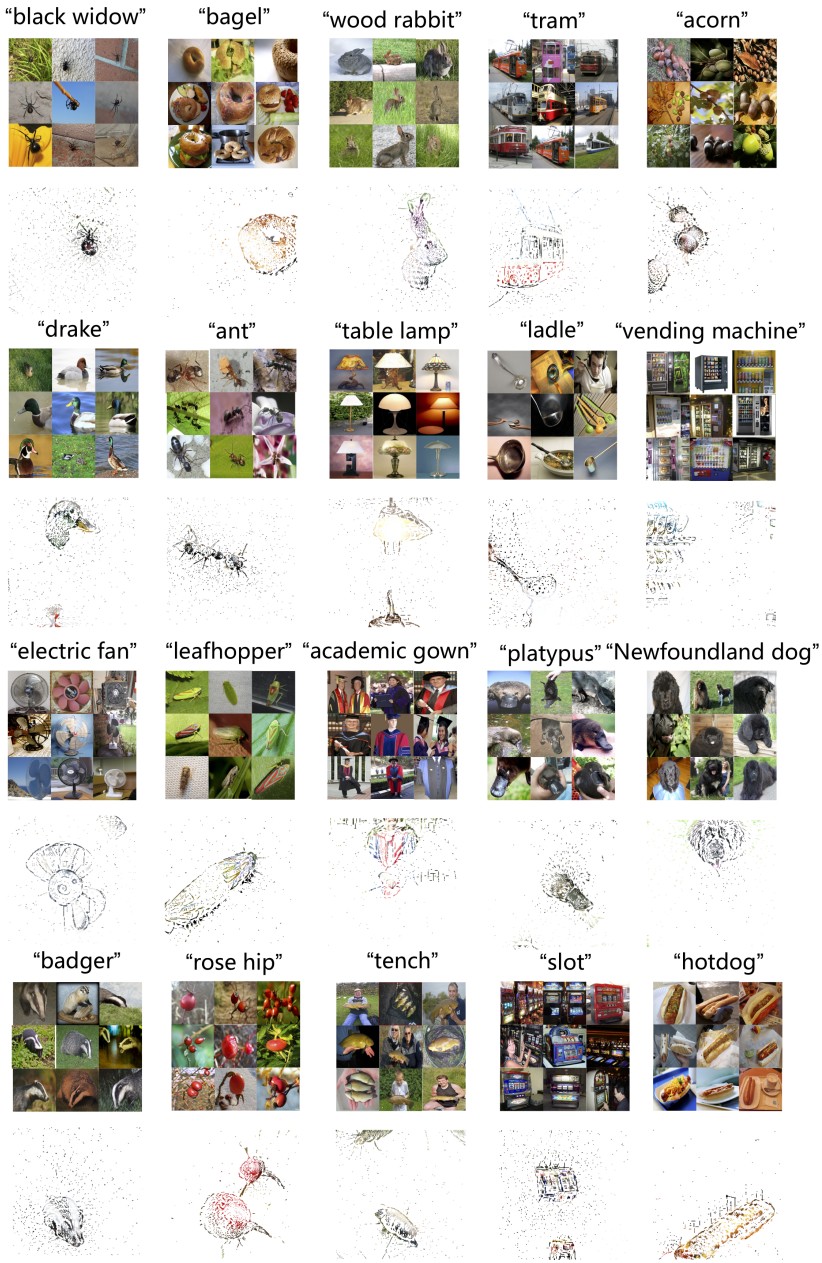

Figure 10: Class-wise patterns found by our method on ImageNet at pattern size 5%. We only show the pattern for 20 out of the 50 tested ImageNet classes. The predictive powers of these patterns are reported in Table 2. Positive canvases are used.

## D.2 PATTERNS REVEALED ON DIFFERENT CANVASES

Following the experiment in Section D.1, here we show the class-wise patterns revealed on the four types of canvases: random, positive, negative, and white. For random, negative and white canvas, we show the pattern revealed on one canvas images, while for the positive canvas, we show the pattern revealed on two canvas images (i.e. Positive1 and Positive2).

Figure 13 illustrates the patterns revealed for 3 CIFAR-10 classes. As can be observed, the patterns found on different canvases share common shapes, although there are some variations in the details and locations of the shape. This confirms that the patterns revealed by our method represent some

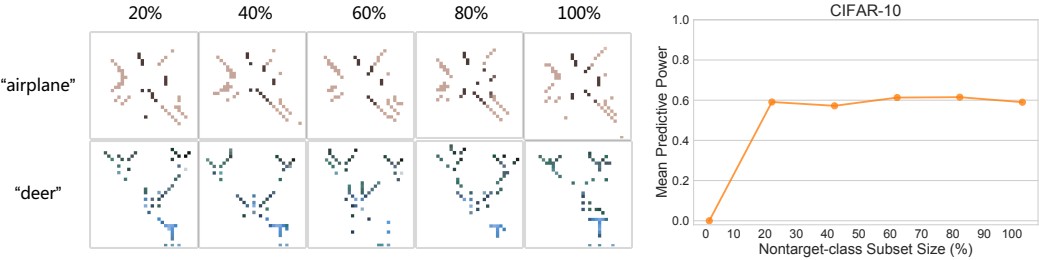

Figure 11: Left grid: Patterns found by our methods under different nontarget-class subset size (% to the test set), for "airplane" and "deer" classes in CIFAR-10. Right plot: The mean predictive power of the class-wise patterns on CIFAR-10. The pattern size is fixed to 5% of the image size.

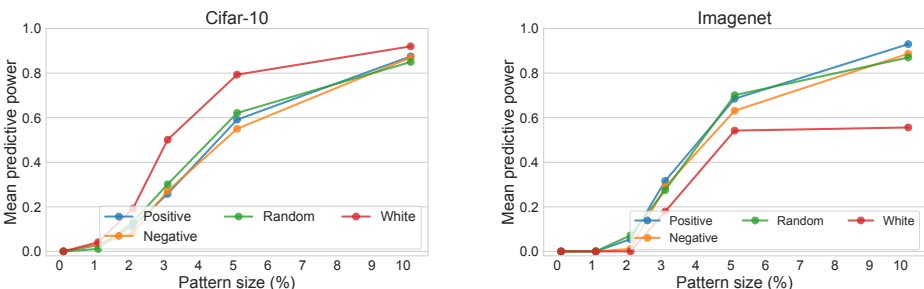

Figure 12: Mean predictive power (mPW) of the class-wise patterns found by our method over all 10 CIFAR-10 classes or 50 randomly selected ImageNet classes for naturally trained ResNet-50 models. The mPW is shown for each of the 4 sampling strategies: random, positive, negative and white. The pattern size varies from 0% to 10% of the image size.

consistent information learned by the model. This also implies that the model may learn to be sensitive to those patterns at different locations of the input space. In other words, DNNs tend to learn certain class-wise patterns at *multiple* (rather than one) locations of the input space. On the other hand, this also means that the shape features exist in one pattern may not be the only pattern learned by the model. Note that the texture (e.g. color) of the pattern is highly correlated to the canvas image, which we have discussed in Section 4.1.

Figure 14 shows the patterns found on ImageNet. For some of the canvases, the revealed pattern has a noticeable correlation with the object in the canvas, especially for the positive canvases. For example, the first two patterns of the "analog clock" class. Note that the patterns revealed by the white canvas also have clear shape features, though there are no canvas dependent features. The shapes in white-canvas patterns tend to appear at more locations of the input space, which may be related to the spatial distributions of the objects in the class.

## D.3 EFFECT OF DIFFERENT POSITIVE CANVASES ON PREDICTIVE POWER

Here, we further show how different *positive* canvases affect the predictive power. We test positive canvases selected from the top-$N$ most confident and correctly classified images. We reveal the patterns learned by ResNet-50 models trained on natural (clean) CIFAR-10 and ImageNet datasets. Same as in Section 4.1 and Appendix D.1, here we search for the pattern on 20% of the test images and test its predictive power on the full test set. The results are reported in Table 4 and Table 5, where it shows that patterns found on the top-1 positive canvas are not necessary the most predictive ones. As we have shown in Figure 13 and 14, different sampling strategies can reveal similar shapes, but the details and the locations of the shapes can be quite different. This is also why we suggest to use $N$ (i.e. $N$=5) different canvases to help find the most predictive patterns.

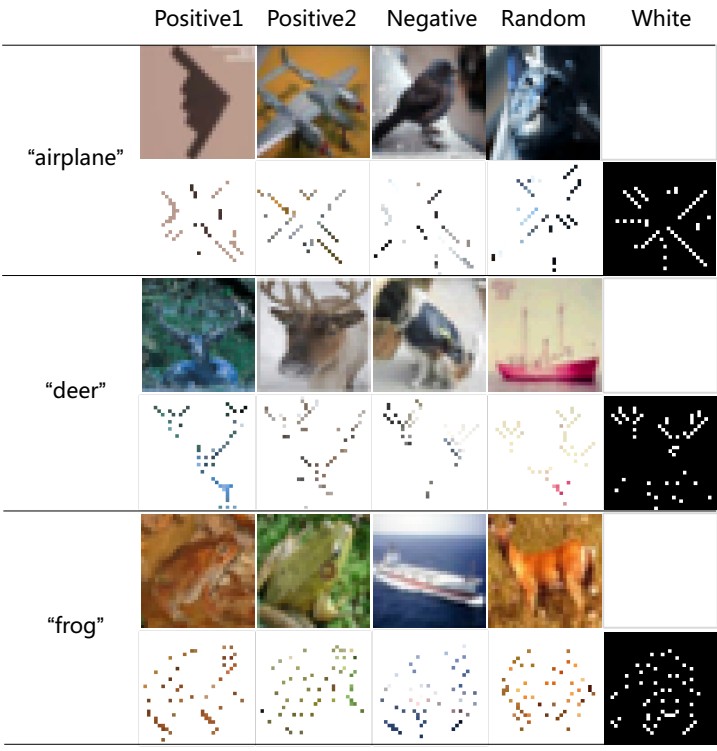

Figure 13: Class-wise patterns revealed by our method using different canvases (e.g. two positive, one negative, one random and one white canvases) on CIFAR-10 for a naturally trained ResNet-50. The pattern size is fixed to 5% of the image size. Here we only show 3 classes as an example.

## E   MORE COMPARISONS TO UNIVERSAL ADVERSARIAL PERTURBATION

Here, we visualize more patterns found by the universal adversarial perturbation (UAP), our method and the TV variant of our method (defined in Equation 3). We use naturally trained ResNet-50 models on clean CIFAR-10 and ImageNet. For each dataset and model, the patterns are generated based on 20% of the test images and tested on the full test set. For UAP, we use the targeted version and set the target class to the class to reveal. The UAP is accumulated in a class-wise manner to generated the pattern separately for each target class. For our methods, the pattern size is set to 5% of the image size. UAP will perturb the entire image, which corresponds to the 100% image size. UAP is bounded by the maximum perturbation $\epsilon \in [8, 32]$. For the TV variant of our method, we set parameter $\beta = 0.2$ or $\beta = 1$ (see Equation 3).

Figure 15 illustrates the class-wise patterns revealed by different methods. The UAP patterns generated on CIFAR-10 share some similarities with our method. However, our method can find much cleaner patterns. The UAP patterns generated on ImageNet are more abstract than CIFAR-10 patterns. This is because UAP explores the *unlearned* space of DNNs, and the *unlearned* space expands with the increase of the input dimensionality (Simon-Gabriel et al., 2018). By contrast, our method is a searching (rather than perturbing) method that does not rely on adversarial perturbations. Therefore, it needs to find the optimal pixel locations in the input space for the pattern to be predictive of the class. This forces the searching algorithm to find the *well-learned* positions of the input space. We believe UAP can be applied in a more controlled manner to explore the well-learned (rather than unlearned) space of DNNs. We find that the TV regularization can help produce smooth patterns with less background noise, however, the main patterns are generally the same as those found without using TV.

Table 6 reports the the mean predictive power of the patterns revealed by the above three methods. The mean predictive power is computed over all 10 CIFAR-10 classes or 50 randomly selected ImageNet classes. For UAP, we use the attack success rate (ASR) as the predictive power. As can be inferred from Table 6, the predictive powers of our method are similar to UAP with maximum perturbation $\epsilon=8$. UAP has much higher predictive power at $\epsilon=16$ or $\epsilon=32$, as expected. For the TV

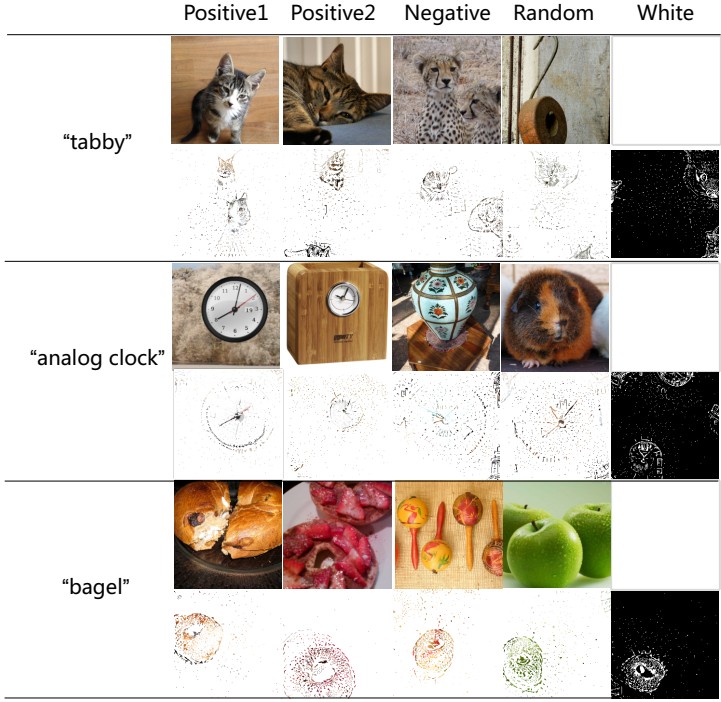

Figure 14: Class-wise patterns revealed by our method using different canvases (e.g. two positive, one negative, one random and one white canvases) on ImageNet for a naturally trained ResNet-50. The pattern size is fixed to 5% of the image size. Here we only show 3 classes as an example.

version of our method, increasing the regularization strength from $\beta$=0.2 to $\beta$=1 tends to find less predictive patterns. This is because TV removes the predictive background noise. This confirms that background noise also plays an important role in DNNs (Ilyas et al., 2019), which also aligns with our findings in the adversarial setting in Section 4.4.

## F   MORE PATTERNS REVEALED ON BACKDOORED MODELS

In addition to our analysis in Section 4.3, here we test two more types of BadNets triggers and apply our method to reveal these triggers. We train a ResNet-50 on BadNets poisoned CIFAR-10 set to obtain the backdoored ResNet-50, i.e., 10% of the data is poisoned following (Gu et al., 2017). We set the backdoor target class to "airplane", and use three types of backdoor triggers (shown in Figure 16). The backdoored models are trained without using any data augmentations. This is a typical setting for backdoor attack as data augmentation will make it more difficult for the model to remember the trigger pattern (Gu et al., 2017). As a comparison, here we also reveal the patterns for backdoored models trained with data augmentations: random shift, random crop and random horizontal flip.

Figure 16 illustrates the patterns revealed by our method at pattern size 1% (of the image size). As shown in the figure, our method can reliably reveal the backdoor trigger pattern learned by the model, for backdoored models trained with or without data augmentations. In the no data augmentation case, the recovered trigger patterns are very close to the ground truth triggers. This indicates that DNNs can memorize both the trigger pattern and its location. Interestingly, in some of the recovered patterns, there are also trigger-similar patterns at different places of the input space. For example, the middle two columns in right subfigure of Figure 16. This indicates that the model may learn trigger-similar patterns from natural data, as we discussed in 4.3. As illustrated in the left panel of Figure 16, data augmentations tend to shift the trigger location.

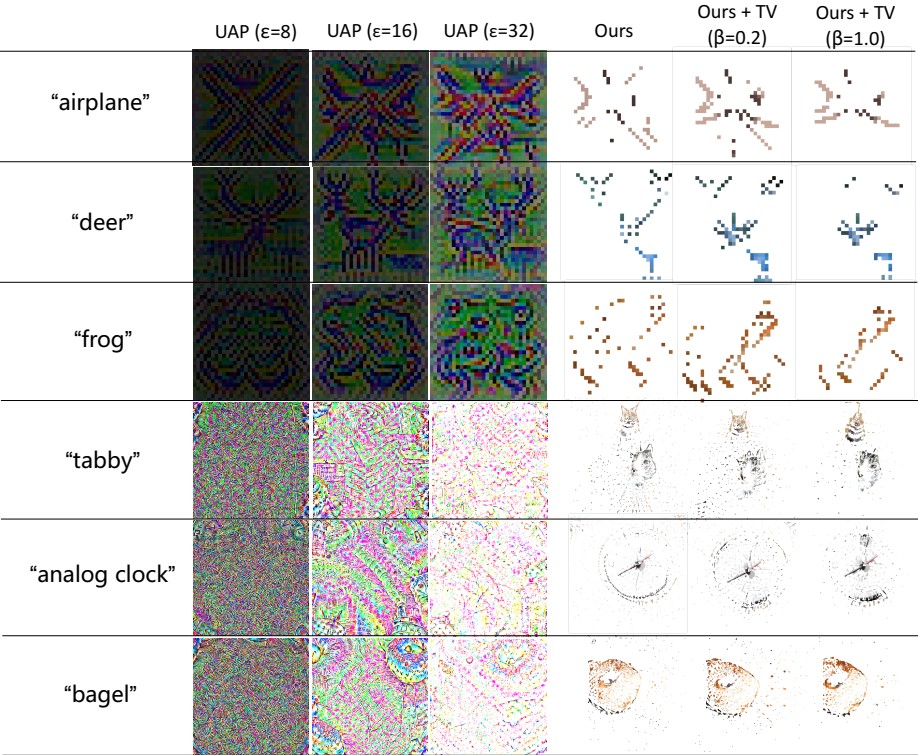

Figure 15: A comparison of the class-wise patterns revealed for naturally trained ResNet-50 models by universal adversarial perturbation (UAP), our method and our method with the total-variation (TV) (Fong & Vedaldi, 2017) regularization (defined in Equation 3). For our methods, positive canvas is used and the pattern size is set to 5% of the image size. UAP perturbs the entire (i.e. 100%) image. $\epsilon$ parameter is the maximum perturbation constraint for UAP, while $\beta$ is the parameter for TV regularization. The top and bottom 3 rows show the patterns for 3 CIFAR-10 and ImageNet classes, respectively.

## G  ATTENTION SHIFT WHEN A CLASS-WISE PATTERN IS ATTACHED

We use the attention map to check the network's attention shift when the class-wise pattern found by out method is attached to an image. As shown in Figure 17, we find that class-wise pattern has very strong attention around the areas that contain clear shapes, and the attention of the network is significantly shifted towards those areas when the pattern is attached to different images. This confirms that the patterns identified by our method are indeed universal patterns that carry consistent information.

## H  EXTENSION TO LEARNED CANVAS

Here we test an extension of our method to simultaneously perturb the mask $\boldsymbol{m}$ and the canvas $\boldsymbol{x}_c$ as follows:

$$\min_{\boldsymbol{x}_c, \boldsymbol{m}} \log f_y(\boldsymbol{m} * \boldsymbol{x}_c + (\tilde{1} - \boldsymbol{m}) * \boldsymbol{x}_n) + \alpha \frac{1}{n} \|\boldsymbol{m}\|_1 . \qquad (4)$$

This is a direct extension of our method defined in Equation 2, with the canvas $\boldsymbol{x}_c$ also being perturbed during the searching process. This extension can also be interpreted as a combination of our mask perturbation in Equation 2 and the universal adversarial perturbations (UAP) applied on the canvas image $\boldsymbol{x}_c$. Note that, different from UAP, here the perturbation on the canvas is not bounded. By solving Equation 4, the canvas is also learned to be predictive, which might potentially lead to more precise and predictive patterns.

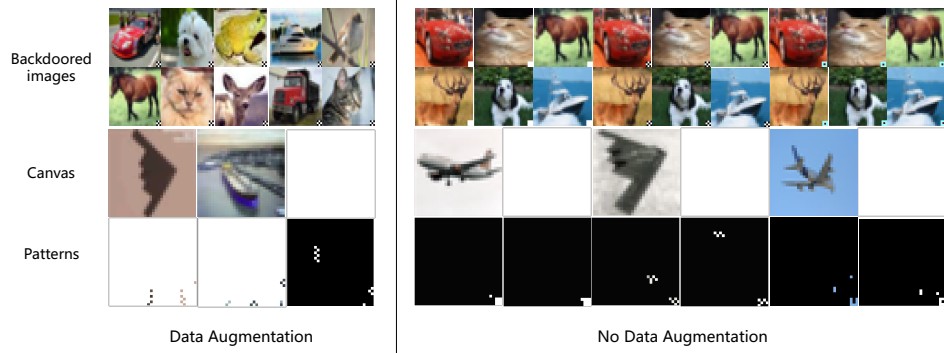

Figure 16: Backdoor patterns revealed by our method for different backdoor triggers. The backdoored ResNet-50 model is trained on CIFAR-10 and poisoned by BadNets towards the target class "airplane". The pattern size is set to 1% of the image size, and the patterns are searched based on 20% of the test set. Both positive and white canvases are used. Data augmentation means the backdoored model was trained with data augmentation techniques. For the data augmentation setting, we also use the negative canvas (the middle column in the left panel).

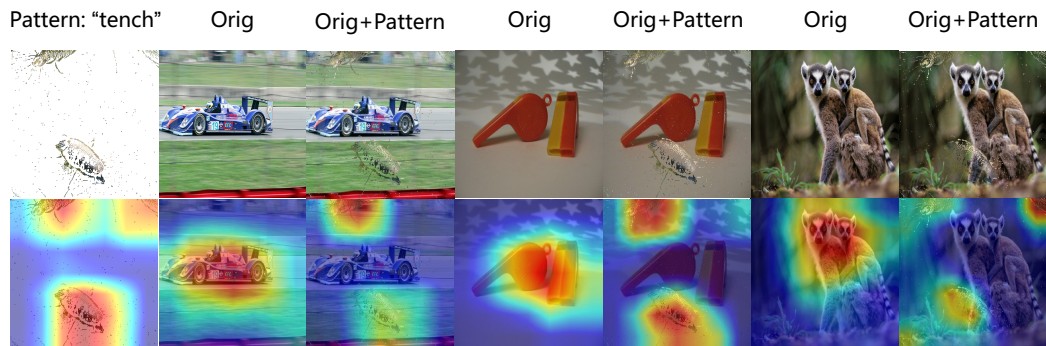

Figure 17: The attention (visualized by Grad-CAM) shift of a ResNet-50 model (on ImageNet) when our class-wise pattern is attached to different images. The pattern is from "n01440764"("tench")

In the above objective, there are two different types of variables that need to be optimized at the same time: the mask $m$ and the canvas $x_c$. To tackle this, we apply an alternating optimization strategy as follows. We update the mask and the canvas each for 5 steps alternatively with step size $2/255$ for the canvas and step size $0.02$ for the mask. The same normalization and clipping techniques are used here as in our original method. We use positive sampling to select the initial canvas image and set the pattern size to 5% of the image size. The patterns are searched for based on 20% of the test set images. We compare side-by-side the patterns revealed by the two methods in Figure 18.

On CIFAR-10 dataset, the patterns found on learned canvas also contain shape features, but for some classes, it tends to focus on only the most important part of the object. For example, the antlers found for the "deer" class. This is because, by perturbing the canvas, the method will explore the most vulnerable part of the canvas that can lead to minimized loss towards the target class. On ImageNet, perturbing the canvas tends to find patterns at the four corners or edges of the canvas image, as shown in the right panel of 18. This is because those regions are the most under-learned and vulnerable space that can most effectively make the network to predict the target class. Overall, perturbing the canvas tends to force the method to explore the most vulnerable and unlearned feature space of DNNs.

The predictive powers of the patterns found on fixed versus learned canvases are reported in Table 7. The predictive power is tested on the entire test set of either CIFAR-10 or ImageNet. Overall, the predictive powers of the patterns found on learned canvases are similar to those found on fixed canvases, although there are some variations. Surprisingly, the learned (perturbed) canvas does not lead to more predictive patterns, considering the high effectiveness of adversarial perturbations. Upon

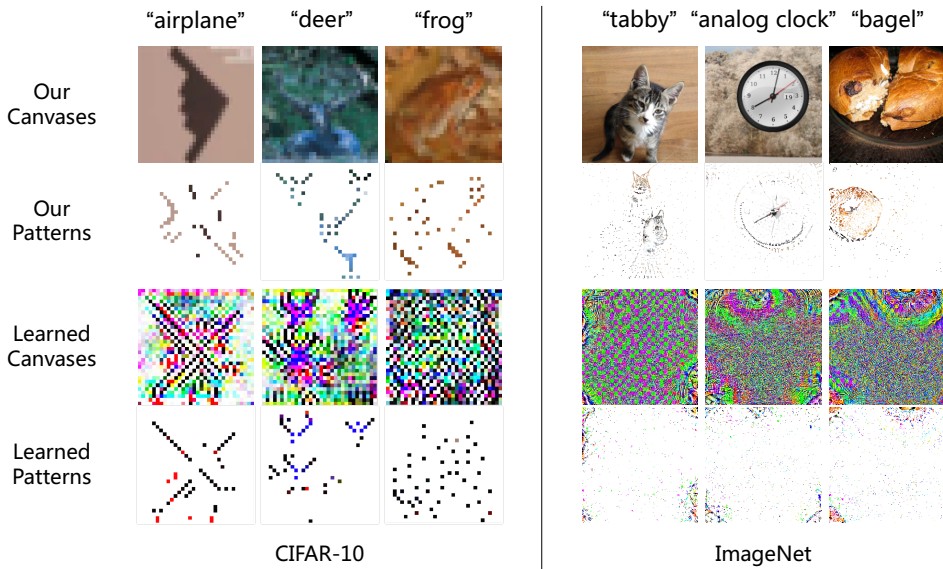

Figure 18: A comparison of the class-wise patterns revealed by our method and an learned canvas version of our method (defined in Equation 4) for naturally trained ResNet-50s on CIFAR-10 and ImageNet. The pattern size is fixed to 5% of the image size. The first row shows the patterns found by the original method, while the learned canvas and the revealed patterns are shown in the second and third rows respectively.

further investigation, we find that this is due to three reasons: 1) the pattern revealed when using the learned canvas is a class-wise universal adversarial pattern, which is less effective (predictive) than a sample-wise adversarial perturbation; 2) only 5% of the perturbed canvas is used to extract the pattern, which significantly reduces the adversarial (predictive) effect; 3) the pattern needs to be attached to other clean images to compute the predictive power. This is a transfer scenario that also reduces the adversarial effect. This eventually results in similarly predictive patterns for both the fixed canvas and the learned canvas.

Table 2: Detailed predictive power of different sizes of the patterns found by our method for ResNet-50 trained on natural ImageNet. The predictive power is shown separately for each of the 50 ImageNet classes. **mPW**: mean predictive over all 50 classes; **STD**: standard deviation over all 50 classes. Positive canvases are used.

| Class | Pattern Size | | | | | Original Acc |
| --- | --- | --- | --- | --- | --- | --- |
| | 1% | 2% | 3% | 5% | 10% | |
| tench | 0.000 | 0.054 | 0.293 | 0.609 | 0.641 | 0.920 |
| black widow | 0.000 | 0.116 | 0.661 | 0.768 | 0.779 | 0.860 |
| drake | 0.000 | 0.083 | 0.523 | 0.821 | 0.952 | 0.840 |
| platypus | 0.000 | 0.000 | 0.013 | 0.275 | 0.750 | 0.800 |
| dowitcher | 0.000 | 0.062 | 0.300 | 0.656 | 0.856 | 0.900 |
| oystercatcher | 0.000 | 0.007 | 0.177 | 0.469 | 0.677 | 0.960 |
| chesapeake bay retriever | 0.000 | 0.057 | 0.193 | 0.636 | 0.852 | 0.880 |
| schipperke | 0.000 | 0.036 | 0.155 | 0.667 | 0.929 | 0.840 |
| newfoundland dog | 0.000 | 0.183 | 0.695 | 0.878 | 0.902 | 0.820 |
| toy poodle | 0.000 | 0.146 | 0.667 | 1.270 | 1.292 | 0.480 |
| leopard | 0.000 | 0.091 | 0.625 | 0.705 | 0.659 | 0.880 |
| ant | 0.000 | 0.105 | 0.776 | 0.921 | 0.987 | 0.760 |
| leafhopper | 0.000 | 0.044 | 0.378 | 0.656 | 0.778 | 0.900 |
| wood rabbit | 0.000 | 0.000 | 0.070 | 0.465 | 0.802 | 0.860 |
| badger | 0.000 | 0.000 | 0.081 | 0.453 | 0.674 | 0.860 |
| gorilla | 0.000 | 0.013 | 0.113 | 0.563 | 0.675 | 0.800 |
| academic gown | 0.000 | 0.000 | 0.026 | 0.632 | 1.289 | 0.380 |
| backpack | 0.000 | 0.000 | 0.045 | 0.955 | 1.841 | 0.440 |
| bicycle-built-for-two | 0.000 | 0.081 | 0.302 | 0.698 | 0.814 | 0.860 |
| bookcase | 0.000 | 0.000 | 0.015 | 0.409 | 0.788 | 0.660 |
| castle | 0.000 | 0.025 | 0.163 | 0.538 | 0.750 | 0.800 |
| chain | 0.000 | 0.154 | 1.731 | 2.385 | 2.654 | 0.260 |
| church | 0.000 | 0.029 | 0.353 | 0.912 | 1.088 | 0.680 |
| cradle | 0.000 | 0.000 | 0.100 | 0.800 | 1.925 | 0.400 |
| electric fan | 0.000 | 0.000 | 0.273 | 0.580 | 0.625 | 0.880 |
| go-kart | 0.000 | 0.000 | 0.011 | 0.319 | 0.702 | 0.940 |
| holster | 0.000 | 0.013 | 0.066 | 0.395 | 0.658 | 0.760 |
| ladle | 0.000 | 0.081 | 0.291 | 0.639 | 0.663 | 0.860 |
| lifeboat | 0.000 | 0.117 | 0.596 | 0.766 | 0.702 | 0.940 |
| loupe | 0.000 | 0.130 | 0.761 | 1.391 | 1.717 | 0.460 |
| paper towel | 0.000 | 0.000 | 0.068 | 0.405 | 0.703 | 0.740 |
| ping-pong ball | 0.000 | 0.133 | 0.389 | 0.667 | 0.711 | 0.900 |
| punching bag | 0.000 | 0.214 | 0.600 | 0.914 | 1.000 | 0.700 |
| saltshaker | 0.000 | 0.106 | 0.576 | 0.742 | 0.788 | 0.660 |
| sax | 0.000 | 0.103 | 0.515 | 0.632 | 0.588 | 0.680 |
| slot | 0.000 | 0.117 | 0.511 | 0.713 | 0.713 | 0.940 |
| spotlight | 0.000 | 0.033 | 0.133 | 1.033 | 1.633 | 0.300 |
| teapot | 0.000 | 0.150 | 0.625 | 0.886 | 0.913 | 0.800 |
| tram | 0.000 | 0.020 | 0.200 | 1.000 | 1.260 | 0.500 |
| table lamp | 0.000 | 0.012 | 0.105 | 0.360 | 0.674 | 0.860 |
| vending machine | 0.000 | 0.011 | 0.076 | 0.050 | 0.674 | 0.920 |
| wool | 0.000 | 0.000 | 0.075 | 0.525 | 1.350 | 0.400 |
| traffic light | 0.000 | 0.057 | 0.341 | 0.670 | 0.841 | 0.880 |
| ice lolly | 0.000 | 0.045 | 0.341 | 0.727 | 0.807 | 0.880 |
| bagel | 0.000 | 0.029 | 0.286 | 1.000 | 1.214 | 0.700 |
| hotdog | 0.000 | 0.000 | 0.012 | 0.267 | 0.616 | 0.860 |
| spaghetti squash | 0.000 | 0.000 | 0.012 | 0.159 | 0.671 | 0.820 |
| acorn | 0.000 | 0.104 | 0.531 | 0.739 | 0.792 | 0.960 |
| rose hip | 0.000 | 0.065 | 0.261 | 0.696 | 0.935 | 0.920 |
| coral fungus | 0.000 | 0.000 | 0.010 | 0.188 | 0.521 | 0.960 |
| **mPW** | 0.000 | 0.056 | 0.322 | 0.701 | 0.936 | 0.761 |
| **STD** | 0.000 | 0.057 | 0.308 | 0.347 | 0.405 | 0.188 |

Table 3: Predictive power tested on the 80% test images that are not used for pattern searching. The results in the brackets are predictive powers tested on the entire (i.e. 100%) test set. The results are shown for ResNet-50 models trained on natural CIFAR-10 and ImageNet, and averaged (i.e. mean predictive power) over all 10 classes of CIFAR-10 and 50 randomly selected ImageNet classes.

| Dataset | Pattern Size | | | | |
|---|---|---|---|---|---|
| | 1% | 2% | 3% | 5% | 10% |
| CIFAR-10 | 0.028 (0.028) | 0.112 (0.113) | 0.256 (0.258) | 0.599 (0.591) | 0.861 (0.875) |
| ImageNet | 0.000 (0.000) | 0.067 (0.055) | 0.322 (0.316) | 0.680 (0.685) | 0.932 (0.930) |

Table 4: Predictive power of class-wise patterns revealed by different positive canvases on a naturally trained ResNet-50 on CIFAR-10. The canvases are selected using positive sampling based on the top-$N$ ($N$=5) most confident and correctly classified images. Pattern size is fixed to 5% image size. **STD**: the standard deviation of the predictive power over all 5 canvases. For each class, the predictive power of the best canvas is highlight in **bold**.

| Class | Positive Canvas | | | | | STD |
|---|---|---|---|---|---|---|
| | top-1 | top-2 | top-3 | top-4 | top-5 | |
| airplane | 0.473 | 0.651 | **0.745** | 0.729 | 0.392 | 0.141 |
| bird | **0.665** | 0.623 | 0.470 | 0.220 | 0.591 | 0.161 |
| dear | 0.622 | **0.698** | 0.659 | 0.511 | 0.553 | 0.068 |
| frog | 0.135 | **0.383** | 0.377 | 0.246 | 0.276 | 0.092 |
| Ship | 0.312 | 0.262 | 0.507 | 0.550 | **0.589** | 0.132 |

Table 5: Predictive power of class-wise patterns revealed by different positive canvases on a naturally trained ResNet-50 on ImageNet. The canvases are selected using positive sampling based on the top-$N$ ($N$=5) most confident and correctly classified images. Pattern size is fixed to 5% image size. **STD**: the standard deviation of the predictive power over all 5 canvases. For each class, the predictive power of the best canvas is highlight in **bold**.

| Class | Positive Canvas | | | | | STD |
|---|---|---|---|---|---|---|
| | top-1 | top-2 | top-3 | top-4 | top-5 | |
| tench | 0.599 | **0.609** | 0.433 | 0.420 | 0.476 | 0.081 |
| tibetan terrier | 0.760 | 0.793 | 0.707 | **0.861** | 0.682 | 0.064 |
| academic gown | 0.411 | 0.572 | 0.301 | **0.632** | 0.596 | 0.126 |
| hook | **1.271** | 0.823 | 0.511 | 0.591 | 0.615 | 0.274 |
| Slot | **0.713** | 0.702 | 0.590 | 0.655 | 0.652 | 0.044 |

Table 6: Attack success rate and predictive power of class-wise patterns generated by universal adversarial perturbation (UAP) (Moosavi-Dezfooli et al., 2017) and our method. For our method, we also test the use of the total-variation (TV) (Fong & Vedaldi, 2017) regularization along with our $L_1$ regularization (this variant of our method is defined in Equation 3). The UAP pattern size is 100% of the image size while our patterns ($L_1$ or TV) are 5% of the image size.

| Methods | UAP ($\epsilon$=8) | UAP ($\epsilon$=16) | UAP ($\epsilon$=32) | Ours | Ours+TV ($\beta$=0.2) | Ours+TV ($\beta$=1) |
|---|---|---|---|---|---|---|
| CIFAR-10 | 0.621 | 0.967 | 0.982 | 0.609 | 0.580 | 0.302 |
| ImageNet | 0.607 | 0.683 | 0.944 | 0.623 | 0.597 | 0.272 |

Table 7: Predictive power of class-wise patterns revealed by our method and by the extended version of our method (defined in Equation 4) for naturally trained ResNet-50s on CIFAR-10 and ImageNet. Pattern size is fixed to 5% image size.

| Method | CIFAR-10 Class | | | ImageNet Class | | |
|---|---|---|---|---|---|---|
| | airplane | deer | frog | tabby | analog clock | bagel |
| Fixed Canvas | 0.745 | 0.698 | 0.383 | 0.547 | 0.711 | 1.000 |
| Learned Canvas | 0.760 | 0.659 | 0.515 | 0.164 | 0.667 | 0.816 |

