# OpenReview forum: "What Do Deep Nets Learn? Class-wise Patterns Revealed in the Input Space"
_ICLR.cc/2021/Conference — Reject_

### Official Review · AnonReviewer1 · 2020-10-21
**Insightful and a pleasure to read**

**Rating:** 7
**Confidence:** 4

**Review:**

The papers proposes a simple method for visualizing the patterns learned by deep neural networks in the supervised classification setting. Informally, suppose you have an image x that is "representative" of the class y and let X be a set of images that belong to other classes. The authors propose an optimization problem that looks for a mask (i.e. set of pixels) along with values of those pixels such that when this pattern is added to any image in X, the model will predict the new image to have the label y. This optimization problem can be solved using iterative thresholding and one may control the level of sparsity as the authors studied. Despite its simplicity, it can reveal clear patterns, particularly on high resolution images, such as ImageNet. The authors, then, show how this method can be used to interpret neural networks, detect backdoor attacks during training, and verify robustness.


Overall, it is a great paper, insightful, and a pleasure to read.

Some comments:

1- The authors insist throughout the paper that this is a method for discovering class-wise patterns. To me, it appears to be a sample-wise pattern since a single representative sample x is used. When multiple samples are used, multiple patterns are detected at different locations as shown in some of the figures. I don't understand why the authors make such an emphasis. The algorithm seems great for providing interpretability per sample. Perhaps, this is because other methods for sample-wise interpretability already exist, such as attention maps, but I don't think that is a serious issue since it does not hurt to have multiple approaches for the same problem. I would suggest that the authors mention, at least, in the paper that this can be used sample-wise as well for interpretability.

2- I am curious to know if upsampling CIFAR10 to have 224x224 images with a reasonable interpolation method would generate better patterns (closer to the patterns observed in ImageNet).

3- In Section 4.3, the authors speculate that augmentation is responsible for the fact that the method finds the trigger patterns at different locations. This is easy to verify by simply training without augmentation. Has this been done?

4- It would be better for readability if all images are placed at the top of the pages, rather than in arbitrary places.

---

> ### Author Response · Authors · 2020-11-23
> **Response to AnonReviewer1:**
>
> Thanks for your thoughtful comments. Please find our responses below.
>
> ---
> **Q1:** Sample-wise vs class-wise method.
>
> **A1:** We agree with you that a sample-wise interpretation of our method wouldn’t hurt our contribution. It is a sample-wise method from the canvas perspective, however, the patterns found by our method reveal more class-wise information. As shown in Figure 14, for some of the canvases, the revealed pattern has a noticeable correlation with the canvas, especially for the positive canvases. This implies that the patterns found by our method do carry sample-wise features from the canvas image. Meanwhile, they are also very predictive of the entire class, as when applied on non-target class images, the model predicts the target class. This is the reason why we emphasize that the patterns are class-wise. We also find that different sampling strategies can reveal similar patterns, and more importantly, the patterns revealed by the white canvas also contain clear shapes, though there are no sample-wise features.
>
> ---
> **Q2:** Can upsampling CIFAR10 to 224x224 images would generate better patterns?
>
> **A2:** This is a very interesting suggestion. Actually, some of the shapes are quite abstract for the CIFAR-10 classes. Generating better patterns on CIFAR-10 by upsampling may be a feasible and powerful way. However, the main focus of our work is to show the patterns learned by DNN in typical (i.e. not upsampled) settings.  We will conduct the suggested analysis in our future works.
>
> ---
> **Q3:** Empirical verification of the impact of data augmentation regarding Section 4.3 and Figure 6.
>
> **A3:** Thanks for the suggestion. We have verified this in the revision. We have run the experiment without any data augmentation and updated  the new results to Figure 7 (left subfigure). We have added more visualizations to Figure 16 and Appendix F. We have also adjusted our analysis.
>
> ---
> **Q4:** It would be better for readability if all images are placed at the top of the pages, rather than in arbitrary places.
>
> **A4:** Thanks for the suggestion. We have moved the images to the top of the pages.
>
> ---

---

### Official Review · AnonReviewer4 · 2020-10-27
**Interesting idea but needs a different focus**

**Rating:** 4
**Confidence:** 5

**Review:**

#### Summary
This work proposes a technique to compute sparse patterns in the input space that are highly predictive of a class, even when added to samples from a different class (non-target samples).
Patterns for class $c$ are produced by optimizing a point-wise weighted average of two samples (a fixed sample called the canvas and one that changes after each optimization step referred to as $x_s \in X_s$) s.t. the result is highly predictive (yields a low error) for $c$. Learned weights for the average (the mask) are thresholded and used as a binary mask for the canvas (which is either a sample that has been labeled/predicted as class $c$ or a white image).
Analysis of the resulting patterns focuses on explainability on various fronts: first, the consistency of the pattern is evaluated (via a custom metric called 'predictive power') which shows that resulting patterns do cause a CNN to predict the class that has been optimized for.
A few remarks on the patterns are made and also some claims regarding the link to the model's learned features (like position, structure, and number of shapes).
Attention maps confirm that the saliency of the patterns prevails over the original content of the canvas image. Finally, models trained adversarially and with backdoors are analyzed by computing the sparse-mask patterns.


#### Strong and weak points of the paper:
##### Strengths:
 - The paper is easy to read and the main ideas can be followed. A wide array of experiments are conducted with a small and (at least a subset of) a large dataset.
 - The proposed method is straightforward and described in enough detail to be easily reproducible.
 - Comprehensive and relevant related work has been cited and compared.
 - Section 4.2: an interesting baseline showing that the resulting patterns are more effective at causing the model to predict a target class than highly responsive areas of a "natural" sample.
 - Section 4.3 and 4.4: the inclusion of an analysis of training regimes that are known to affect the feature space learned by a neural network (adversarially trained, trained with robust features, and trained with backdoors) provides relevant use-cases for a method that is proposed as an interpretability tool.
 - Section 4.4: interesting results that seem to agree with Ilyas et al. 2019 in that more robust models require a perturbation that looks more like the adversarial target.

##### Weaknesses:
 - The main concern is that the goal of the paper is not aligned with the findings: I consider that the patterns shown are a valuable tool to analyze the behavior of a neural network, but from a completely different perspective. These patterns (and the algorithm to obtain them) suggest that there are global-targeted adversarial perturbations (a close analogous to the work of Moosavi et al.: global perturbation because it is one perturbation applied to a set of samples; targeted because each perturbation has been designed to cause attacked samples to predict one specific class). In turn, the patterns' appearance is as difficult to interpret as those of classical adversarial perturbations or global perturbations. Claims regarding shape, texture, position, etc, are not well supported by measurable evidence (beyond a few observational remarks supported by a handful of examples), and are more likely the result of the method by which these perturbations are obtained (e.g, the regularization w.r.t. the mask, clipping, binarization, point-wise multiplication). Instead of trying to interpret the appearance of such perturbations, a focus on the implications of global-targeted adversarial perturbations can provide valuable insights into the manifold's structure learned by modern neural networks as well as their shortcomings.
 - Most experiments are missing key details regarding their setup. In particular, the canvas image is often not shown (especially relevant for Figures 1, 2, and 7). From figures where it is shown (like Figure 4), it seems like regardless of the canvas, a binary pattern with a global-targeted perturbation is learned.
 - An analysis of the impact of the canvas is missing w.r.t. predictive power which could support/invalidate the idea of the learned pattern to be a global-targeted adversarial perturbation.
 - Section 4.1, Patterns Revealed by Positive Canvas: several claims regarding the properties of the patterns (e.g., structural consistency, number, and position of shapes) in the mask are only supported by a qualitative evaluation of a few samples. No metrics are proposed.
 - Section 4.1, Patterns Revealed by Positive Canvas: although a metric (predictive power) is used to measure the consistency of the pattern w.r.t. a target class, results are based on a few classes. Why not report it for all 10 classes on CIFAR10? Also, results on 5 classes of Imagenet is not representative of the variety found in that dataset. A simple average of the predictive power over each class would provide stronger support for the results found in the subset of selected classes.
 - Section 4.1, Patterns Revealed by Different Canvases: the absence of texture in the patterns is more likely caused by the sparsity term on equation 2 (together with the threshold and the point-wise multiplication with the canvas) rather than by an intrinsic property learned by the model.


#### Recommendation:
 - In its current form and with the proposed focus, I consider that this paper cannot be accepted. The focus of the analysis can yield stronger, more conclusive results from the standpoint of global adversarial perturbations.

#### Supporting arguments for recommendation:
 - Without the right context (i.e., not that of explainability but adversarial attacks) the analysis of what the patterns represent is poorly supported by experiments. There is a lot of work done in the direction of adversarial attacks that findings in this work could re-purpose. Meanwhile, the impact of proposing a "global-targeted adversarial attack" based on one sample seems promising, but it is out of the scope of this work's analysis. Refer to the list of weaknesses for more details.

#### Questions to clarify:
 - How are the class-patterns related to universal perturbations?
 - How is this method fundamentally different (i.e., more advantageous) than a universal adversarial perturbation (Moosavi-Dezfooli 2017) generated for each class?
 - If the learned pattern says something about the (clean) feature space of the classifier, what do these patterns reveal about classes with a large intra-class variation? For example, the class "clock" contains digital and analog clock faces (on the extreme, digital clocks show only numbers, and analog ones only the hands of the clock). Apart from their semantic relationship, there is no visual correspondence between the two sub-categories. Would this scenario be a limitation of the proposed method or what does it reveal about the model/the data?
 - Section 3, Equation 2: what happens to the mask (therefore to the conclusions from the patterns in the mask) with other regularization metrics like Total Variation (TV)? TV is used in [2] to find masks with connected areas that are representative for each sample in a counterfactual manner (i.e., blurring the area of the mask, causes the model to predict something else). In fact, not using TV makes the mask converge to an adversarial perturbation. Comparing those findings to those on this work (in particular Equation 2), makes me think that the resulting mask is more of a targeted adversarial attack rather than a representative pattern of the class. Moreover similar work published in ICLR 2016 [3] shows how almost any source image can be 'anchored' to the prediction of a guide image (equivalent to the canvas image in this work). This paper is essentially computing a sparse version of that attack via a masked input and the output layer (in [3] they don't apply masks and the deep representation was not constrained to the output but an arbitrary layer). It would be interesting to show the reconstruction of samples that have been "attacked" with the patterns in this work (following [3]).
 - Section 3, sampling canvas and results in Figure 4: how is the predicted power affected by the choice of canvas (positive, negative, or white)? Evaluating on a few classes can be misleading and although the metric of predictive power (PW) seems adequate, I suggest presenting a global metric by taking the mean PW (mPW) to better support the generality of the corresponding results.
 - Experiments with Imagenet: can you specify which classes the chosen IDs map to? Are they all related somehow? What were the criteria to select those 5? Why only 5?
 - Figure 2: it would be important to see what the canvas image looks like in order to establish how much of the pattern is just coming from that canvas.
 - Section 4.1, Patterns revealed by Positive Canvas: how are the number of shapes measured? Where does one shape start and one ends? It seems this observation is rather empirical and only a few samples are shown as evidence (one may ask how often does this happen and whether it happens because the dataset also has multiple shapes -which requires no mask, just simply looking at the samples). The same issues arise with claims about the position where the patterns occur (how is this measured?).
 - Section 4.1, Patterns Revealed by Different Canvases: how is the consistency of patterns evaluated? Is there further evidence beyond the 4 samples shown in Figure 4 for CIFAR10? Those seem to share structural similarity but, is this always the case? How do you establish structural similarity on the samples from Imagenet?
 - Section 4.2: taking one test image per class (source) whose highest activation area (with GradCAM) is pasted into ...how many target images? In other words, are the results in figure 5 (right) based on how many samples overall? This experiment needs to be described in more detail to be reproducible.
 - Claims for backdoor images about the trigger being recoverable is rather weak for two reasons: (1) there is only one class being evaluated and more importantly (2) there is only one (simple) pattern that has been analyzed. From this experiment, there are some indications that backdoor patterns may be reliably retrieved but nothing conclusive (the full trigger pattern has not been reconstructed and other, more complex patterns have not been evaluated).


#### Additional feedback (not necessarily part of the decision assessment):
 - The notion of 'backdoor' is not explained in the abstract and it is not clear from the context what it could be. Including a brief description about it on the abstract could improve readability.
 - In the second paragraph of Section 1: references for "backdoor" attacks and "adversarial" attacks are swapped.
 - Section 1, 4th paragraph: the notion of "canvas image" is used without it being defined.
 - Section 2, "General Understandings of DNNs": there is work that has also focused on modeling properties of the input space over the dataset (and not only sample-wise) [1].
 - Section 3 paragraph "Canvas Sampling": typo for "reveal".


#### References:
[1] Palacio, Sebastian, et al. "What do deep networks like to see?." Proceedings of the IEEE Conference on Computer Vision and Pattern Recognition. 2018.

[2] Fong, Ruth C., and Andrea Vedaldi. "Interpretable explanations of black boxes by meaningful perturbation." Proceedings of the IEEE International Conference on Computer Vision. 2017.

[3] Sabour, Sara, et al. "Adversarial manipulation of deep representations." ICLR (2016).

---

> ### Author Response · Authors · 2020-11-23
> **Response to AnonReviewer4 - Part III**
>
> ---
> **Q12:** Figure 2: it would be important to see what the canvas image looks like in order to establish how much of the pattern is just coming from that canvas.
>
> **A12:** We have included these results to Figure 13 (for CIFAR-10) and Figure 14 (for ImageNet), showing patterns for all 4 types of canvases. For some of the canvases, the revealed pattern indeed has a correlation with the canvas, the patterns found on different canvases share common shapes, although there are some variations in the details and locations of the shape. The analysis has been added to Section 4.1 and Appendix D.2.
>
> ---
> **Q13:** How are the number of shapes measured? Where does one shape start and one ends? It seems this observation is rather empirical and only a few samples are shown as evidence.
>
> **A13:** The reviewer is correct that our observations about shapes are empirical. Like we explained before, quantitative measures for the shapes are very difficult to design. We have thought about using some Euclidean distances or inception distances like FID/IS (for GANs), it is still hard to fairly determine the threshold for “similar” versus “dissimilar”. This type of issue is also frequently encountered in the field of interpretable/explainable machine learning.
>
> ---
> **Q14:** Section 4.2: taking one test image per class (source) whose highest activation area (with GradCAM) is pasted into ...how many target images? In other words, are the results in figure 5 (right) based on how many samples overall?
>
> **A14:** We use Grad-CAM to extract the high attention area of the positively sampled canvas image (a test image that is correctly classified and has the highest confidence) for each of the 50 ImageNet target classes (class names are in Table 2). The size of the high attention area is set to be 10% of the image. We then attach the extracted high attention area to all non-target class test images to compute its predictive power (the same testing scheme as for our method). The **mean** predictive power over the 50 target classes has been updated to Figure 5 (right plot). We have also included these details in Section 4.2 “Comparison to Attention Map”.
>
> ---
> **Q15:** Claims for backdoor images about the trigger being recoverable is rather weak.
>
> **A15:** We have updated the result of mPW of all the 10 classes of CIFAR-10 in Figure 7, for the backdoored model. We apply BadNets to backdoor attack and train a backdoored model for each class. We then average the PW tested on each pattern. We have also added the results against two more BadNets trigger patterns in Figure 16 and Appendix F.
>
> ---
> **Q16:** Additional feedback.
>
> **A16:** we have fixed these issues in the update.
>
> ---
> **References:**
>
> [1] Christian Szegedy, Wojciech Zaremba, Ilya Sutskever, Joan Bruna, Dumitru Erhan, Ian Goodfellow, and Rob Fergus. Intriguing properties of neural networks. International Conference on Learning Representations (ICLR), 2014.
>
> [2] Xingjun Ma, Bo Li, Yisen Wang, Sarah M Erfani, Sudanthi Wijewickrema, Grant Schoenebeck, Dawn Song, Michael E Houle, and James Bailey. Characterizing adversarial subspaces using local intrinsic dimensionality. In ICLR, 2018.
>
> ---

---

> ### Author Response · Authors · 2020-11-23
> **Response to AnonReviewer4 - Part II**
>
> ---
> **Q7:** Section 4.1, Why not report the patterns for all 10 classes on CIFAR10? Also, results on 5 classes of Imagenet are not representative of the variety found in that dataset. A simple average of the predictive power over each class would provide stronger support for the results found in the subset of selected classes.
>
> **A7:** Thanks for the suggestion. We have added the pattern visualizations for all CIFAR-10 classes and 20 ImageNet classes in Figure 9 and Figure 10, respectively. We have also updated the predictive power to the suggested mean predictive power over all CIFAR-10 classes and 50 randomly selected ImageNet classes in Figure 3 and Figure 12. We have also the detailed predictive power for each class in Table 1 and Table 2. The analysis of the results have also been adjusted in the related sections or appendices.
>
> ---
> **Q8:**: Section 4.1, Patterns Revealed by Different Canvases: the absence of texture in the patterns is more likely caused by the sparsity term in equation 2 (together with the threshold and the pointwise multiplication with the canvas) rather than by an intrinsic property learned by the model.
>
> **A8:** The absence of texture in the patterns is indeed partially caused by the sparsity controls. But we would like to point out that these sparsity controls are to find a few number of pixel positions in the input space that are predictive of the class. However, this regularization does not restrict the pixels in the pattern to be texture or shape. It is an intrinsic property of DNNs because the pixels in the pattern are distributed in a way that naturally forms some interesting shapes rather than a wild pattern. We have added this discussion to Section 4.1 “Patterns Revealed by Positive Canvas”.
>
> ---
> **Q9:** If the learned pattern says something about the (clean) feature space of the classifier, what do these patterns reveal about classes with a large intra-class variation? For example, the class "clock" contains digital and analog clock faces (on the extreme, digital clocks show only numbers, and analog ones only the hands of the clock). Apart from their semantic relationship, there is no visual correspondence between the two sub-categories. Would this scenario be a limitation of the proposed method or what does it reveal about the model/the data?
>
> **A9:** The pattern revealed by our method is highly predictive, however, this does not mean it is the only pattern learned by DNNs. We have added this analysis to Appendix D.2. As shown in Figure 10 for 20 ImageNet classes, DNNs learn more than one shape at different locations of the input space, especially on some of the relatively large intra-class variation classes. We will leave the visualization of more diverse ImageNet classes (e.g. the super classes) to our future work.  In general, this will be an issue of knowledge encoding, i.e. choosing class labels for a database that are not overly specific and not overly general.  Whilst this is an important issue, our method assumes such a knowledge encoding has already been undertaken.   If the encoding is effective, our method is likely to produce useful results.  If the encoding is poor, then our method will be less likely to be useful results.  However, in this latter case the visualisations of our method could potentially help guide the user to updating their knowledge encoding.
>
> ---
> **Q10:** What happens to the mask with other regularization metrics like Total Variation (TV)? Relation to [3], and what the reconstructions look like?
>
> **A10:** Thanks for the suggestion. We have run additional experiments using the TV regularization in Section 4.2 (Figure 6) and Appendix E (Figure 15). Overall, we find that the TV regularization does help to produce smoother patterns, however, the general patterns are similar to that without using TV. We have also included an analysis of TV on the predictive power in Table 6 (Appendix E). In general, TV has a negative impact on the predictive power: more TV regularization tends to produce smoother patterns, but lower predictive power.
>
> While the reconstruction suggestion is interesting, we are not able to complete this experiment within the time limit. We believe the reconstructed images will be different from the **canvas** images, at least for the negative, random and white canvases. This is because, when the pattern (extracted from the canvas) is attached to an image, the model will predict the target class, not the class of canvas image.
>
> ---
> **Q11:** Experiments with Imagenet: can you specify which classes the chosen IDs map to? Are they all related somehow? What were the criteria to select those 5? Why only 5?
>
> **A11:** We have updated Figure 3 to include the mean predictive power results for all 10 CIFAR-10 and 50 randomly selected ImageNet classes. We also visualized the patterns for all CIFAR-10 and 20 ImageNet classes in Figure 9 and 10 (Appendix C.1).
>
> ---

---

> ### Author Response · Authors · 2020-11-23
> **Response to AnonReviewer4 - Part I**
>
> Thanks for your thoughtful comments. We have included a set of new experimental results to clarify some of your concerns. Please also find the following responses.
>
> ---
> **Q1:** Correlation and difference to universarial adversarial perturbation (UAP).
>
> **A1:** We agree with you that the patterns identified by our method behave like targeted universal adversarial perturbations: they are universal across multiple samples and predictive of a target class. However, the two methods work in different ways. By fooling the network, UAP explores the **unlearned** space (low-probability ``pockets") of the network [1,2]. In contrast, our method is a searching (rather than perturbing) method that does not rely on adversarial perturbations. Thus, it has to find the optimal pixel locations in the input space that are **well-learned** by the model for the pattern to be predictive of the class. We have added a discussion subsection “Difference to Universal Adversarial Perturbation” to Section 3.
>
> ---
> **Q2:** Advantage of our method over UAP.
>
> **A2:** We have added an empirical comparison of the patterns found by our method and those found by UAP, in Section 4.2 “Comparison to Universal Adversarial Perturbations”. Different to UAP, our method focuses more on the locations of the pixels that can form a predictive pattern. As such, the patterns found by our method are cleaner, and less abstract, especially on ImageNet (Figure 6). More visual comparisons are provided in Appendix E. As we mentioned in “Why is it Class-wise?” in Section 3, we believe UAP can also be applied to serve our purpose, however, it should be applied in a more controlled manner so as to explore the well-learned (rather than unlearned) space of DNNs.
>
> ---
> **Q3:**  The patterns' appearance is as difficult to interpret as those of classical adversarial perturbations or global perturbations.
>
> **A3:** We agree that interpretability can be somewhat subjective. However, it is very challenging to reveal the knowledge learned by DNNs in a meaningful and human-interpretable form. This is why we propose to visualize the patterns in the pixel space. To help the interpretation, we have visualized more patterns for all CIFAR-10 classes and 20 ImageNet classes in Figure 9 and 10, respectively. We have also included a visual comparison of the patterns revealed by UAP and our method in Figure 6, Section 4.2. We agree that our method can still be improved to find even more interpretable patterns, for instance, the use of a color palette. We will leave this to our future work.
>
> ---
> **Q4:** Most experiments are missing key details regarding their setup. In particular, the canvas image is often not shown.
>
> **A4:** We have added the canvases and the revealed patterns in Figure 13 for CIFAR-10 and Figure 14 for ImageNet, and an analysis in Appendix D.2.
>
> ---
> **Q5:** An analysis of the impact of the canvas is missing w.r.t. predictive power which could support/invalidate the idea of the learned pattern to be a global-targeted adversarial perturbation.
>
> **A5:** We have added the predictive power (PW) with respect to different canvases in Appendix D.1 and D.3 (Figure 12, Table 4, and Table 5). As shown in Figure 12, the predictive powers of the patterns revealed on positive, negative and random canvases are similar. Note that, for these three types of canvases, our method utilizes $N$ (N=5) canvas images and selects the most predictive pattern (out of the 5 patterns found on the 5 canvas images) as the final pattern for PW testing. Different to these three types of canvases, white canvas exhibits a quite distinctive impact: higher PW on CIFAR-10, yet lower PW on ImageNet. Within the **same type** of canvases (e.g. positive canvas), the predictive power varies on different canvas images (see Table 4 and Table 5).
>
> ---
> **Q6:** Section 4.1, Patterns Revealed by Positive Canvas: several claims regarding the properties of the patterns (e.g., structural consistency, number, and position of shapes) in the mask are only supported by a qualitative evaluation of a few samples. No metrics are proposed.
>
> **A6:** We agree that quantitative metrics will definitely make our results more convincing. However, it is difficult to design such measures. This type of issue is also frequently encountered in the field of interpretable/explainable machine learning.  We would like to include more qualitative results if the reviewer can give us some suggestions.
>
> ---

---

### Official Review · AnonReviewer2 · 2020-10-28

**Rating:** 6
**Confidence:** 4

**Review:**

This paper proposes a visualization method to reveal the class-specific discriminative patterns of DNNs in the input space. When added to images from another class, such patterns can lead the DNN to classify the images into the pattern's class. From the experimental results, the authors conjecture that images trained on natural data can have backdoors. It also claims that the method reveals the trigger patterns of backdoor attacks, that adversarially trained models learn simplified shape patterns, but an intentionally-perturbed robust dataset improves model robustness by sacrificing its ability to represent shapes.

Despite feeling unsure about certain points, I think this paper is well-written. However, some of its main conclusions do not seem to be well-supported by the experimental results and analysis.

Strengths:
1. Proposing the interesting idea and method to find class-wise patterns that change the DNNs' prediction into a certain target image when added to images from another class. Such patterns are shown to reveal the abstract shapes and texture of the target class.

2. Generalizing the analysis to various scenarios including backdoored models and adversarially-robust models.

Weaknesses:
1. The conjecture that "DNNs trained on natural data can also have backdoors" does not seem to be well-supported by the "predictive power" in Figure 3, due to the definition of predictive power. My understanding of a backdoor trigger for image classification is some pattern that generalizes well to images that are not used to craft such triggers. However, by definition of predictive power, it is evaluated on the examples that are used to generate such patterns, since $ACC(f(x_n+p_y), y)$ seems to be defined on the nontarget-class images. This is more like finding universal targeted adversarial attacks in the white-box setting, which may not be so surprising to succeed. I think it would be much more convincing if $ACC(f(x_n+p_y), y)$ is defined on a different set of nontarget-class images that are not used to craft the patterns.

2. It is shown in Figure 6 (right) that the patterns from backdoored models do not transfer well to natural models, and it is suggested that such a method can be used to detect backdoored models. However, the results for transferability across natural models trained from different initializations seems to be missing. We do not know how the pattern's predictive power changes on another natural model that is not used to craft the pattern, and if it drops significantly, we cannot expect to use the method to detect backdoored models.

3. Patterns shown in Figure 6 (left) do not seem to reveal the trigger at all. Only the pattern from the white canvas match the pattern, but its position is not correct. It says the position mismatch may be caused by data augmentation, but it is not convincing enough since no results on backdoored model trained without data augmentation is shown. It would be much better if such results can be included.

4. The patterns from Figure 7 are also not conclusive enough. For example, patterns for the "deer" class seems to be in a worse shape when crafted from $\hat{\mathcal{D}}_R$ than ${\mathcal{D}}$, but it is the opposite for the "airplane" class. Therefore, I do not think we can conclude that "intentionally-perturbed dataset improves model robustness by sacrificing its ability to
represent shapes".


### Updates after the rebuttal
I appreciate the authors' timely response. The latest update explaining the regularization effect of using a fixed canvas $x_c$ and a learnable mask to avoid overfitting to spurious features, which seems to be inevitable if using a learnable $x_c$, does seem plausible and differentiates itself from the general idea of UAT. From the updated content in Appendix H, using a learnable $x_c$ does not seem to be able to find interpretable patterns. I feel contributing a new method with some improvements is worth an accept.

However, I still feel the results are not conclusive enough. Regardless of the final decision, I hope to see more comparisons with simpler baselines in the future version. To highlight the novelty and effectiveness of the proposed method, the authors should try to compare with more baseline approaches that do not have presumptions on $x_c$ and initialize $x_c$ randomly. Currently, $x_c$ are selected as images having highest prediction scores from the model. Instead, we can learn $x_c$ from random initializations by using some approaches that enhance the transferability of adversarial examples. There are simple methods on improving the transferability of adversarial examples, e.g., adding Gaussian noise when crafting adversarial examples [1], or adding different random data augmentations [2]. These methods may also lead to interpretable patterns but the effectiveness of the resulting patterns could be stronger. I do not fully agree that a desirable canvas has to be a neutral or unbiased image.

[1] Wu, Lei, Zhanxing Zhu, Cheng Tai, and Weinan E. "Understanding and enhancing the transferability of adversarial examples." arXiv preprint arXiv:1802.09707 (2018).
[2] Huang, Qian, Isay Katsman, Horace He, Zeqi Gu, Serge Belongie, and Ser-Nam Lim. "Enhancing adversarial example transferability with an intermediate level attack."  CVPR (2019).

---

> ### Author Response · Authors · 2020-11-23
> **Response to AnonReviewer2:**
>
> Thanks for your valuable comments. Please find our responses below.
>
> ---
> **Q1:** Definition of predictive power on the examples that are used to generate such patterns.
>
> **A1:** In our setting, the patterns were searched based on 20% of the test data, and their predictive powers were tested on the entire (i.e. 100%) test set. To make it clearer, we have updated the definition of PW and how it is computed in Section 4 “Predictive Power”. We have also added an analysis of the predictive power tested on the 80% of test data that were not used for pattern searching in Appendix B.2. The predictive power results are generally the same.
>
> ---
> **Q2:** This is more like finding universal targeted adversarial attacks in the white-box setting, which may not be so surprising to succeed.
>
> **A2:** UAP is a perturbation method, while our method is a searching (rather than perturbing) method that does not rely on adversarial perturbations. Thus, it has to find the optimal pixel locations in the input space that are **well-learned** by the model for the pattern to be predictive of the class. We have added a more detailed discussion of the difference of our method to UAP in Section 3 “Difference to Universal Adversarial Perturbation”. We have also added an empirical comparison to UAP in Section 4.2 “Comparison to Universal Adversarial Perturbations”, as well as Appendix E. Please also find more clarifications in our response **A1** to AnonReviewer4.
>
> ---
> **Q3:** In Figure 6 (right) that the patterns from backdoored models do not transfer well to natural models. How can this help backdoor detection?
>
> **A3:** We have added the transfer study in Section 4.3 (Figure 7, right subfigure). The patterns revealed by our method transfer well across separately trained DNNs. Additionally, we have updated the middle subfigure of Figure 7 and show the same size (1% image size) of **natural** is not predictive on **naturally trained** models. Therefore, the existence of a small highly predictive pattern can indeed indicate that a backdoor trigger has been learned.
>
> ---
> **Q4:** Patterns shown in Figure 6 (left) do not seem to reveal the trigger at all. The effect of data augmentation needs further verification.
>
> **A4:** Thanks for the suggestion. We have updated Figure 7 (left subfigure) to show the patterns revealed without data augmentation, and more visualizations in Figure 16 in Appendix F. The new result confirms our previous analysis. We have also adjusted our claims in the revision.
>
> ---
> **Q5:** The patterns from Figure 7 are also not conclusive enough.
>
> **A5:** We agree that our previous claim is not precise. We find that the shape distortion by $D_{R}$ is rather class-dependent. The patterns learned on $D_{R}$ reveal that the "robust" perturbation can help remove the background noise (e.g. the "airplane" class).  However, it may also lose a certain part of the robust shape (e.g. the body of the "deer"). To better illustrate the effectiveness of our method in adversarial settings, we have included visualizations of more classes and adjusted our analysis and claims in Section 4.4.
>
> ---

---

> > ### Comment · AnonReviewer2 · 2020-11-24
> > **Further questions**
> >
> > Thanks for the careful rebuttal, and provide new experimental results comparing with universal adversarial training (UAT). After a more careful read, I realized that previously I did not realize the canvas image $x_c$ is from the dataset, instead of learned. This distinguishes the proposed method from UAT in that UAT does not use canvas images from the dataset to generate the adversarial perturbations.
> >
> > However, I think the authors should justify why using $x_c$ from the dataset is better than learning $x_c$ directly. The latter one is different from the original objective in the UAT papers, but it should still be counted as a UAT algorithm. Specifically, I hope the authors could justify the following objective is not as good as the proposed canvas approach in discovering the backdoors or fooling the networks (target attacks):
> >
> > $\min_{x_c,m} -\log f_y(m*x_c+(1-m)*x_n) + \alpha \frac{1}{n}\lVert m \rVert_1.$
> >
> > It might be too late to raise such concerns but I still want to see the comparisons.

---

> > > ### Author Response · Authors · 2020-11-25
> > > **Our Response:**
> > >
> > > Thanks for your interesting suggestion. We consider the suggested method to be different to UAP: UAP is a univariate optimization problem while the suggested method is a bivariate (e.g. optimizing two types of variables: the mask and the canvas, at the same time) optimization problem which requires quite different optimization strategy. We have run the suggested method using alternating optimization to guarantee convergence, and updated the new results to Appendix H.
> > >
> > > We find that the learned canvas tends to explore the most vulnerable and under-learned space of DNNs that can most effectively fool the model to predict the target class. Consequently, it finds patterns at the four corners or the edges of the input space for ImageNet classes (Figure 18 right panel). We also find that the patterns extracted from the learned canvases are not particularly more predictive than our method. We have also provided an analysis of the reasons in Appendix H.
> > >
> > > We were not able to finish the backdoor experiment in the limited time.  However, we believe the learned canvas is exploring a different space to our method.
> > >
> > > We are happy to add more results to our final version if the reviewer has more questions.

---

> > > > ### Comment · AnonReviewer2 · 2020-11-25
> > > > **Thanks for the new results! But I still don't feel convinced.**
> > > >
> > > > Thank you for the hard work! But I think there is some miscommunication here. My question is why you have to choose canvas images from the dataset, or what is the advantage of using a canvas images initialized as some image from the dataset, instead of learning a $x_c$ from random or even zero initializations. You seem to have taken the canvas image for granted without justifying why it is better. I think the justification is important because without such a selection of canvas image, the proposed method does not seem fundamentally different from UAT, and I would assume UAT could achieve the same effect as the proposed method.
> > > >
> > > > In fact, from Figure 4, even using a white image as $x_c$, minimizing Eq. (2) w.r.t. $m$ can reveal similar patterns as using canvas images. This result already seems to indicate that there is no need to use canvas images, and when using the white images, the method does not really seem different from UAT. In your new experiments in Appendix H, you are learning $x_c$ initialized as some image from the dataset, but the patterns turns out less similar to the white image, which is a little bit strange.
> > > >
> > > > To sum up, I do not think every paper has to propose a novel method, but the messages from this paper does not really seem surprising neither. Still, the paper has provided lots of interesting explorations. I will not be against an accept, but I still feel more novel insights are needed.

---

> > > > > ### Author Response · Authors · 2020-11-25
> > > > > **Thanks for the question**
> > > > >
> > > > >
> > > > > Just to clarify, the canvas does not have to be sampled from the dataset, for example, our use of the white-canvas. The canvas plays an important role in our algorithm *as a constraint or regularization on the search space for finding the pattern.* A key insight here is that the canvas should not be an image that has been adversarially perturbed or learned. Identifying the quality and best choice for a canvas is an open issue, but our belief is as follows:
> > > > >
> > > > > **Desirable canvas:** A good canvas should be a neutral or unbiased image.   It can be a natural image or a white image or a random image (a space where all pixels are equally important).   In other words, the canvas should not be a result of some optimization process, especially a process that has been targeted to fool the model.
> > > > >
> > > > > **Undesirable canvas:** If a canvas has been learned or is the result of adversarial perturbation, the resulting pattern may be overfitted or biased towards the most vulnerable region of the input space. I.e. it could be a reflection of the non-robust features [1] learned by the model. The existing universal adversarial perturbation methods do not constrain the perturbation to prevent exploration of the non-robust space.  This is not surprising, since that is what universal adversarial perturbations are designed for (i.e. their primary objective being to fool the model)
> > > > >
> > > > > In a learned canvas, there may be a few pixels that are highly influential for the model’s prediction. Using this type of canvas is more biased, and will end up finding only those few pixels.
> > > > > For example as shown in Figure 18, on ImageNet, use of the learned canvas has resulted in finding a pattern where the influential pixels are (very surprisingly) located at the corners of the pattern images (the bottom row).
> > > > >
> > > > > [1] Adversarial Examples are not Bugs, they are Features.  Ilyas et al.  Neurips 2019.
> > > > > [2] Universal adversarial perturbations. Moosavi-Dezfooli, et al. ICCV 2017.

---

### Official Review · AnonReviewer3 · 2020-10-29
**A simple but reasonable method**

**Rating:** 7
**Confidence:** 4

**Review:**

This work studied what the model has learned from the data. It proposed a class-wise pattern searching method to demonstrate the existence of class-wise predictive patterns in the input space. It studied three training settings, including natural, backdoored and adversarial, and revealed different characteristics of these training settings.

I think that the motivation is very clear,  and the writing is readable, though some minor typos or inconsistencies should be corrected.

The proposed class-wise pattern searching method is very simple but reasonable. My opinion is that it may get somewhat inspiration from the backdoor triggers, since the format of the mixed input in Eq. (1) is same with the poisoned sample with triggers in the backdoor learning. If a discussion about this connection is added, then the derivation will be more smooth.

More details about the minimization of $\mathcal{L}$ in Eq. (2) should be added. Although one algorithm is presented in Appendix A, there is still no sufficient explanations and analysis (e.g., the convergence). And, there are many hyper-parameters in the algorithm. Their settings and adjustments should be more clear, though there are some brief descriptions at the beginning of Section 4.

The studies about three trainings settings appreciated. And the presented results indeed provide some insights of these training settings.

The proposed method reveals many interesting insights of what DNN has learned from the data under different training settings. However, if the insights could be used to improve the current training method or DNNs, the value of this work could be larger. I would like to see some discussions about this point in this manuscript.

---

> ### Author Response · Authors · 2020-11-23
> **Response to AnonReviewer3:**
>
> Thanks for your accurate interpretation of our method and the thoughtful comments. Please find below our responses to your questions.
>
> ---
> **Q1:** Connection to backdoor attack.
>
> **A1:** Thanks for the suggestion. We have added this connection to Section 3, “Motivation and Intuition”.
>
> ---
> **Q2:** More details about the minimization of $\mathcal{L}$ in Eq. (2):  the searching Algorithm, settings and hyper-parameters.
>
> **A2:** We have added a detailed step-by-step description of our algorithm, and  the parameters used in each step in Appendix A. The $\alpha$ parameter (L1 regularization strength) is stated in Section 4 “Experimental Setting”, and the $\gamma$ parameter  (defines the pattern size) is stated with each experiment. Other training hyper-parameters have also been added to Appendix A.
>
> ---
> **Q3:** Insights on how to improve the current training method or DNNs.
>
> **A3:** Thanks for the suggestion. We have added discussions in Section 5. One application of our method is DNN understanding and interpretation. Based on our method, one can develop metrics to measure the strength, weakness or biases (e.g. gender, color and other attributes) of the knowledge learned by DNNs. The other potential application is using our method to develop effective backdoor defense methods by monitoring and avoiding the learning of backdoor triggers during training. Our method can also motivate more effective adversarial defense methods, for example, regularized adversarial training methods that can help DNNs learn more robust shape features.
>
> ---

---

### Author Response · Authors · 2020-11-23
**Rebuttal Summary:**

We sincerely thank all reviewers for their valuable comments and suggestions. The discussions have helped us to greatly improve our paper.  We made the following updates during the rebuttal:

---
* Figure 3: updated the results to mean predictive power (mPW) and mPW under different canvas sampling strategies.
* Figure 5: updated the results to mPW of the attention map.
+ Figure 6: added a comparison between universal adversarial perturbation, our method and our method with total-variance (TV) regularization.
* Figure 7: updated the recovered pattern from the backdoored model trained without data augmentation; updated the results of mPW over all 10 CIFAR-10 classes; added the results of transfer rate on separately trained models.
* Figure 8: added one more example in adversarial settings.
+ Figure 9: added class-wise patterns for more CIFAR-10 images.
+ Figure 10: added class-wise patterns for more ImageNet images.
+ Figure 12: added the results of mPW under different canvas sampling strategies.
+ Figure 13: added class-wise patterns revealed on different canvases on CIFAR-10.
+ Figure 14: added class-wise patterns revealed on different canvases on ImageNet.
+ Figure 15: added more comparison results of universal adversarial perturbation, our method and our method with TV regularization.
+ Figure 16: added backdoored patterns revealed for 2 more backdoor triggers.
+ Table 1: added detailed PW for each of the 10 CIFAR-10 classes.
+ Table 2: added detailed PW for each of the 50 randomly selected ImageNet classes.
+ Table 3: added the results of PW tested on 80% images that are not used for pattern searching.
+ Table 4: added the PW of the patterns revealed on different positive canvases for CIFAR-10.
+ Table 5: added the PW of the patterns revealed on different positive canvases for ImageNet.
+ Table 6: added the predictive power results for universal adversarial perturbation, our method and our method with total-variance (TV) regularization.
* Section 3: added our motivation from backdoor attack; added a discussion on the difference of our method to universarial adversarial perturbation.
* Section 4: added more details of how PW is defined and computed.
* Section 4.1: added more analyses of different types of canvas.
* Section 4.2: added the comparison to universal adversarial perturbation.
* Section 5: added the discussion of the potential applications of our method.
+ Appendix B.1: detailed predictive power for 10 CIFAR-10 and 50 ImageNet classes.
+ Appendix B.2: predictive power results on different test sets.
+ Appendix C.1: class-wise patterns for more CIFAR-10 and ImageNet classes.
+ Appendix D.1: effect of different canvases on the predictive power.
+ Appendix D.2: patterns revealed on different canvases.
+ Appendix D.3: effect of different positive canvases on predictive power.
+ Appendix E: more comparisons to universal adversarial perturbation.
+ Appendix F: more patterns revealed on backdoored models.
+ Appendix H: added an investigation of learned canvas.

---
We have revised our paper according to all the valuable comments and please let us know if there is anything still not clear or any other suggestions.

---

### Decision · Program_Chairs · 2021-01-07
**Final Decision**

**Decision:**

Reject

**Comment:**

The authors propose an algorithm that learns sparse patterns of images that are highly predictive of a target class, even if added to a non-target class. The reviewers agree that the algorithm is novel, is tested on a wide array of experiments, and the paper well written.

Unfortunately, it seems that some of the main claims, such as DNNs trained on clean data "learn abstract shapes along with some texture", resort to qualitative evaluation of the few examples shown in the paper. Furthermore, two reviewers were concerned with how one particular design choice in the algorithm might bias the authors' claims. In particular, pointed out that the patterns learned are highly to the initial canvas used, which is not necessarily strongly motivated.

As these two issues are integral parts of the paper, I hesitate to recommend Acceptance at this point.  That said, the approach looks very promising and I hope the authors continue to pursue this idea.